# Control of the polyamine biosynthesis pathway by G$_2$-quadruplexes

Helen Louise Lightfoot[1], Timo Hagen[1], Antoine Cléry[2,3], Frédéric Hai-Trieu Allain[2], Jonathan Hall[1]*

[1]Department of Chemistry and Applied Biosciences, Institute of Pharmaceutical Sciences, ETH Zurich, Zurich, Switzerland; [2]Department of Biology, Institute of Molecular Biology and Biophysics, ETH Zurich, Zurich, Switzerland; [3]Biomolecular NMR spectroscopy platform, ETH Zurich, Zurich, Switzerland

**Abstract** G-quadruplexes are naturally-occurring structures found in RNAs and DNAs. Regular RNA G-quadruplexes are highly stable due to stacked planar arrangements connected by short loops. However, reports of irregular quadruplex structures are increasing and recent genome-wide studies suggest that they influence gene expression. We have investigated a grouping of G$_2$-motifs in the UTRs of eight genes involved in polyamine biosynthesis, and concluded that several likely form novel metastable RNA G-quadruplexes. We performed a comprehensive biophysical characterization of their properties, comparing them to a reference G-quadruplex. Using cellular assays, together with polyamine-depleting and quadruplex-stabilizing ligands, we discovered how some of these motifs regulate and sense polyamine levels, creating feedback loops during polyamine biosynthesis. Using high-resolution $^1$H-NMR spectroscopy, we demonstrated that a long-looped quadruplex in the *AZIN1* mRNA co-exists in salt-dependent equilibria with a hairpin structure. This study expands the repertoire of regulatory G-quadruplexes and demonstrates how they act in unison to control metabolite homeostasis.
DOI: https://doi.org/10.7554/eLife.36362.001

*For correspondence:
jonathan.hall@pharma.ethz.ch

**Competing interests:** The authors declare that no competing interests exist.

## Introduction

Polyamines (PAs) are small poly-cationic molecules present at millimolar concentrations in cells (*Lightfoot and Hall, 2014*). In mammals, the dominant PAs are spermine and spermidine. PAs play critical roles in many processes but their mechanistic workings are rarely investigated and are poorly understood (*Miller-Fleming et al., 2015*). In cells, PAs bind nucleic acids in two ways: with un-specific interactions where they diffuse relatively freely around the polynucleotide, and with site-specific chelation in defined binding pockets with associated function. Levels of PAs are controlled on multiple levels, including synthesis, inter-conversion and depletion (*Figure 1a*) (*Casero and Pegg, 2009*; *Pegg, 2009*), as well as uptake and efflux (*Abdulhussein and Wallace, 2014*). Furthermore, PA-directed feedback loops operate at RNA and protein levels, whereby low levels of PAs are corrected with increased expression of PA synthesis enzymes and decreased activities of negative regulators (*Miller-Fleming et al., 2015*; *Ivanov et al., 2010*; *Perez-Leal and Merali, 2012*). For instance, several PA synthesis proteins (PSPs) (AZIN1, AMD1, PMOX, ODC1, SAT1) are subject to PA-mediated regulation *via* short open reading frames (uORFs) (*Ivanov et al., 2010*). For example, ODC1 is the limiting factor in PA synthesis (*Miller-Fleming et al., 2015*): it has a highly structured 5'UTR and its translation is strongly affected by PA levels. It is also negatively regulated by binding to OAZ proteins, which themselves are regulated by binding AZIN1. PAs bind to nascent OAZ polypeptides, activating a frameshift needed for production of full-length OAZ (*Ivanov et al., 2010*; *Kurian et al., 2011*).

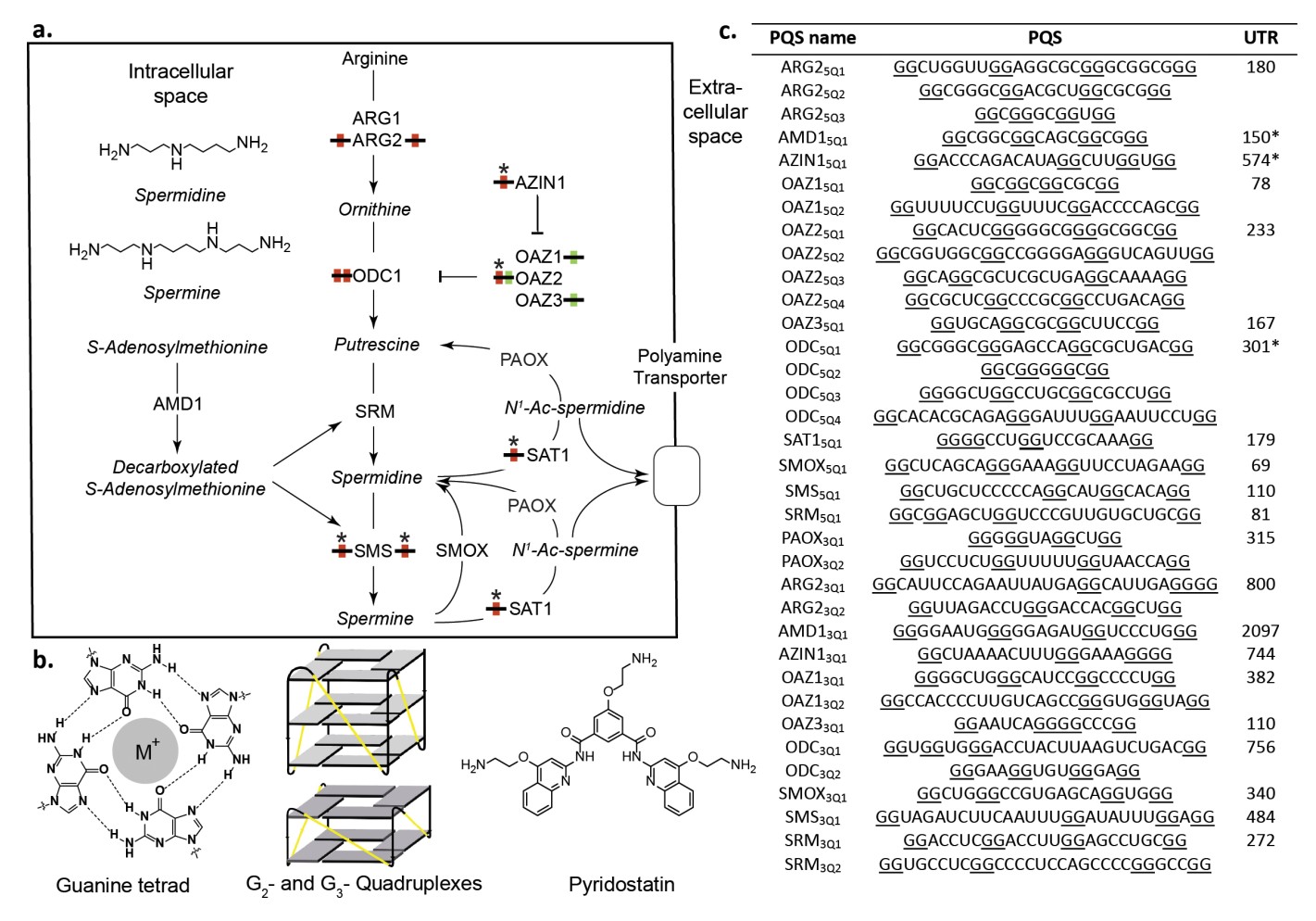

**Figure 1.** Polyamines (PAs) and G-quadruplexes. (a) PA biosynthesis pathway and summary of the roles/regulation of putative quadruplex sequences (PQS's) on PA biosynthesis. Red: $G_2$-rich motifs inhibit PA biosynthesis; Green: $G_2$-rich motifs enhance PA biosynthesis; *: PA-sensing $G_2$-rich motifs. (b) Guanine tetrad stabilized by a monovalent cation ($M^+$) and intramolecular parallel $G_2$- and $G_3$-quadruplexes. Quadruplex loops - yellow. (c) $G_2$-rich motifs in the UTRs of PSPs. PQS's are denoted as the gene name with the UTR and PQS number ($Q_n$; where n≥1). Underlined guanines are those forming the most stable predicted $G_2$-PQS from all possible PQS's predicted by QGRS Mapper (version: Feb. 2014). Full length UTRs were utilized in the study in all cases except for the 5 UTRs of AMD1, AZIN1 and ODC1. *portion of the UTR used in this study for AMD1, AZIN1 and ODC1 (AMD1: 150-AUG; AZIN1: 720-241 and 95-AUG; ODC1: 301-AUG) (See **Supplementary file 1** and **Supplementary file 2** for details). ARG1, ARG2: arginase 1-2; ODC1: ornithine decarboxylase; SRM: spermidine synthase; SMS spermine synthase; AMD1: adenosylmethionine decarboxylase 1; AZIN1: antizyme Inhibitor 1; OAZ1-3: ornithine decarboxylase antizyme 1-3; SAT1: spermidine/spermine N1-acetyltransferase 1; SMOX: spermine oxidase; PAOX: polyamine oxidase.

DOI: https://doi.org/10.7554/eLife.36362.002

A considerable part of gene regulation is controlled by secondary/tertiary structures in UTRs (**Wan et al., 2011**). Examples of structures are stem-loops, pseudoknots and riboswitches, which serve as sensors, reacting rapidly to inputs such as changes in the concentrations of RNA binding proteins (RBPs), metabolites, or even changes in temperature (**Wan et al., 2011**). The intramolecular G-quadruplex consists of stacked guanine tetrads connected by three loops (**Millevoi et al., 2012**) (**Figure 1b**). They self-assemble through Hoogsteen binding and π-π interactions, stabilized by metals or proteins. The stability of G-quadruplexes is governed by the number of G-quartets, the loop length and composition, the flanking nucleotides and salt conditions. Although their structures are difficult to characterize in vivo, G-quadruplexes are recognized as important elements regulating gene expression (**Rhodes and Lipps, 2015**; **Millevoi et al., 2012**), and they are increasingly linked to diseases (**Thandapani et al., 2015**; **Conlon et al., 2016**). Two recent genome-wide studies

identified thousands of such motifs, the majority of which comprised canonical short-looped $G_3$-tracts ($G_3$-quadruplexes) (*Kwok et al., 2016a*; *Guo and Bartel, 2016*). In the latter study, the authors presented evidence that stable $G_3$-quadruplexes (i.e. strong enough to stall reverse transcriptase) in eukaryotic cells were frequently unwound; this suggested the physiological relevance of quadruplex structures should not be automatically inferred from their stability (*Guo and Bartel, 2016*). Consistent with this, several hundred putative metastable RNA $G_2$-quadruplexes have also been predicted throughout the transcriptome (*Kwok et al., 2016a*). So far, few $G_2$-quadruplexes have been studied in detail biophysically, structurally and functionally, all of which carry short loops that are $\leq 7$ nucleotides (nt) in length (*Mullen et al., 2012*; *Morris et al., 2010*; *Cammas et al., 2015*; *Kralovicova et al., 2014*; *Murat et al., 2014*; *Blice-Baum and Mihailescu, 2014*; *Weldon et al., 2017*).

We identified 35 $G_2$-tract putative quadruplex structures (PQS's) in the 5'- and 3'UTRs of genes in the polyamine biosynthesis pathway (PSP). Using cellular reporter assays we showed that twelve of these covering eight PSPs altered reporter activity in comparison to mutants. Strikingly, most of the PQS's increased or reduced reporter expression such that in the setting of their native UTRs they would reduce PA levels. This suggested they might act in unison as regulatory elements to control PA homoeostasis. Using a comprehensive set of independent in vitro biophysical methods, we generated strong supporting data for seven quadruplexes. These included a long-looped conserved quadruplex in the AZIN1 mRNA, which we demonstrated with high-resolution NMR spectroscopy, coexists in a salt-dependent equilibrium with hairpin structures. The activities of four PQS's from *OAZ2*, *AZIN1* and *SMS* correlated with the levels of PAs in cells, suggesting that these $G_2$-PQS's respond to PA levels by an undetermined mechanism(s) in feedback loops. Overall, these findings reveal a previously unrecognized additional mechanism of PA self-regulation involving the entire pathway. We expect that such mechanisms through G-quadruplexes may be a common feature in other metabolic pathways.

## Results

### Predicted $G_2$-PQS's in PSP UTRs

We searched for PQS's in the UTRs of PSPs (*Figure 1a*) using the algorithm QGRS Mapper (*Kikin et al., 2006*), which predicts the ability of a sequence containing G-repeats to fold into (in many cases several) distinct quadruplexes and assigns them a stability score (G-score) based on published biophysical data. Canonical $G_3$-quadruplexes are highly stable in vitro (*Wieland and Hartig, 2007*; *Pandey et al., 2013*) and arguably could be considered as 'thermodynamic sinks' that are unsuitable for fast structural changes during regulation of gene expression. Therefore, to identify more dynamic analogues in PSP UTRs, we applied a weak minimum consensus sequence (G quartets $\geq 2$; total PQS sequence length $\leq 30$) as search criteria. No $G_3$-tract PQS's ($G_3$-PQS: i.e. with the potential to form a $G_3$-quadruplex structure) were found by the algorithm in the UTRs. Intriguingly however, it revealed 20 $G_2$-PQS's across eleven 5'UTRs, and 15 $G_2$-PQS's in ten 3'UTRs of PSPs (*Figure 1c*). Within many of the UTRs multiple overlapping $G_2$-PQS's were predicted and QGRS Mapper highlighted the PQS expected to form the most stable quadruplex structure. It seemed plausible that equilibria might exist between distinct PQS's within these $G_2$-rich regions, possibly influenced by local conditions. To help assess if these $G_2$-containing motifs are functional in cells, the UTR containing the PQS (*Figure 1c*) was cloned 5'- or 3'- to the *Renilla* luciferase coding region in a dual luciferase reporter plasmid. Dual reporter assays are a preferred means (*Halder et al., 2012*) to investigate the function of putative G-quadruplexes in cells. Using full-length UTRs where possible, the properties of a putative regulatory element are investigated by comparison to those of a minimally-mutated control. Thus, we mutated or deleted a minimum number of selected guanines in each PQS so as to prevent the principal PQS, as well as alternative quadruplexes, from forming in the control reporters. During the design we were similarly mindful of recent reports that nucleobases other than G, or even a G-vacancy may substitute for G in a quadruplex (*Švehlová et al., 2016*; *Li et al., 2015*) (*Supplementary file 1*). The influence of each PQS on reporter activity was then assessed by comparison with its respective control after plasmid transfection into HeLa cells.

Eight from 20 PQS's in the 5'UTRs of PSPs affected reporter gene activity. Mutation of PQS's from *ARG2* (ARG2₅Q₁: 40%), *AZIN1* (AZIN1₅Q₁: 49%), *SMS* (SMS₅Q₁: 75%) and *ODC1* (ODC1₅Q₂: 28%; ODC1₅Q₃: 41%) increased *Renilla* activities (*Figure 2a*). Therefore, these PQS's inhibit gene expression and in their natural 5'UTRs would be expected to suppress PA levels. SAT1 and OAZ2 are negative regulators of PA synthesis. In cells, translation of SAT1 is increased in response to high PAs (*Casero and Pegg, 2009*), possibly through an uORF and/or a stem-loop in its coding region (*Perez-Leal and Merali, 2012*) (*Supplementary file 1*). Its most stable predicted quadruplex (SAT1₅Q₁) is conserved in mouse (*Figure 2—source data 1*) and is unusual because it contains no first loop: parallel intramolecular $G_3$-quadruplex structures containing no first loop in DNA were recently described and characterized using a combination of biophysical techniques (*Piazza et al., 2017*). (*Figure 1*). Surprisingly however, we found that mutation of three nucleotides in this PQS

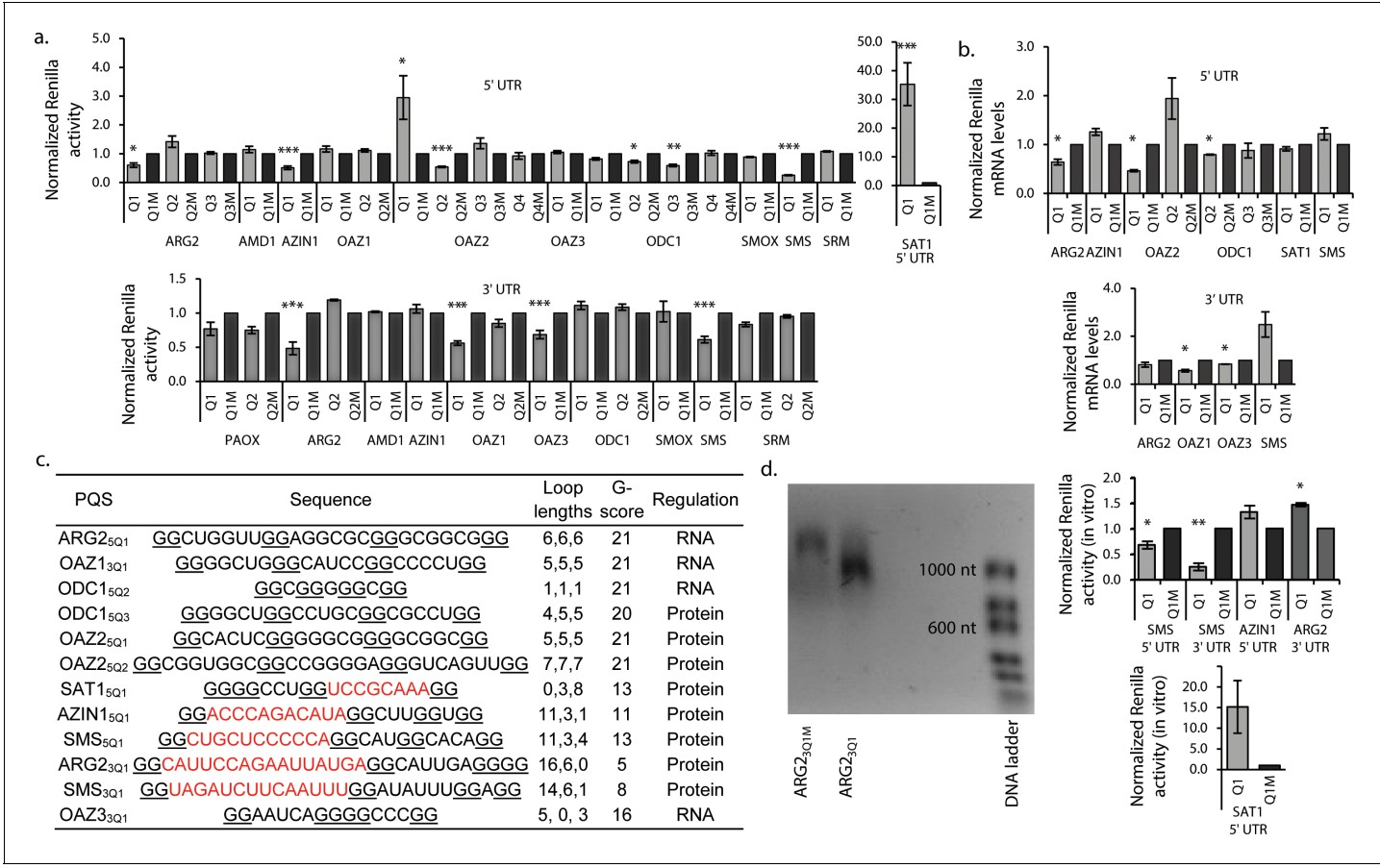

**Figure 2.** $G_2$-PQS's regulate luciferase reporter genes. (**a**) Effect of PQS's from PSP UTRs on *Renilla* luciferase activity in HeLa cells; PQS's ARG2₅Q₁, AZIN1₅Q₁, OAZ2₅Q₁, OAZ2₅Q₂, ODC1₅Q₂, ODC1₅Q₃, SMS₅Q₁, SAT1₅Q₁, ARG2₃Q₁, OAZ1₃Q₁, OAZ3₃Q₁ and SMS₃Q₁ show a statistically significant difference to their mutated controls (n = 3–7; *p≤0.05, **p≤0.01, ***p≤0.001); remaining PQS's are designated as inactive (n>/=2); error bars represent standard error (SE)). (**b**) Effect of functional PQS's on *Renilla* mRNA levels in HeLa cells by qRT-PCR (n = 3–4, *p≤0.05; error bars represent standard error (SE)). (**c**) Properties of the functional $G_2$-PQS's. Underlined: predicted G-tetrad as in *Figure 1c*. Red, bold: long (>7 nt) loops. (**d**) Effects of $G_2$-PQS's on in vitro translation of *Renilla* luciferase in HeLa lysates (n = 3, *p≤0.05, **p≤0.01) (high variability of the SAT1₅Q₁ was noted between different lysate batches). The effect of ARG2₃Q₁ on in vitro transcription of *Renilla* luciferase. (See *Figure 2—figure supplement 1*). Error bars represent standard error (SE). Percentage changes discussed in the text are calculated as differences in normalized *Renilla* counts (x100).

DOI: https://doi.org/10.7554/eLife.36362.003

The following source data and figure supplement are available for figure 2:

**Source data 1.** Conservation of $G_2$-PQS's in PSP UTRs.

DOI: https://doi.org/10.7554/eLife.36362.005

**Figure supplement 1.** PAGE migration of long-looped $G_2$-PQS's in PSP UTRs upon in vitro transcription.

DOI: https://doi.org/10.7554/eLife.36362.004

reduced *Renilla* activity by >30 fold, implying that this $G_2$-PQS contributes strongly to SAT1 regulation. *OAZ2* has four PQS's in its 5'-UTR (*Figure 1c*): mutation of $OAZ2_{5Q1}$ inhibited *Renilla* activity (195%), whereas mutation of $OAZ2_{5Q2}$ enhanced activity (46%). Thus, these PQS's might influence PA synthesis in context-dependent fashion. Mutations in four PQS's in the 3'-UTRs of PSPs increased *Renilla* activities ($ARG2_{3Q1}$: 52%; $OAZ1_{3Q1}$: 44%; $OAZ3_{3Q1}$: 31%; $SMS_{3Q1}$: 39%). Accordingly, in their natural environments PQS's from *ARG2* and *SMS* 3'-UTRs would also act as inhibitors of PA synthesis.

In summary, 12 $G_2$-PQS's from the 5'-UTRs (*ARG2*, *AZIN1*, *OAZ2*, *ODC1*, *SMS*), and the 3'-UTRs (*ARG2*, *SMS*, *OAZ1*, *OAZ3*) altered reporter expression (ten inhibited, two induced). In their native UTRs, nine of these would suppress levels of PAs (summarized in *Figure 1a*), providing a potentially powerful level of structure-based regulation throughout the pathway. Conservation of RNA structure across species is indicative of function, though the study of quadruplex sequence covariation is still in its infancy (*Švehlová et al., 2016*). We therefore assessed the conservation of these PQS's using H-QGRS (*Menendez et al., 2012*), where conservation is measured by factors such as composition, location and predicted stability. Eight $G_2$-PQS's are conserved ($AZIN1_{5Q1}$, $SMS_{5Q1}$, $ODC1_{5Q2}$, $SAT1_{5Q1}$) between mouse and human or primate and human ($SMS_{3Q1}$, $ARG2_{3Q1}$, $ODC1_{5Q3}$ and $OAZ3_{3Q1}$) (*Figure 1*), under highly stringent conservation conditions. We investigated further these 12 motifs with a wide range of assays to provide further insight into their functions.

## Long-looped $G_2$-PQS's in PSPs regulate translation in vitro

Quadruplexes are capable of altering gene expression through their influence on numerous cellular processes including replication, transcription, splicing, mRNA localisation and translation (*Rhodes and Lipps, 2015*). To determine if effects observed in the reporter assays (*Figure 2a*) originated at the mRNA levels, we assayed *Renilla* mRNAs by qRT-PCR. No significant differences were seen in mRNA levels between reporter and their matched controls for $AZIN1_{5Q1}$, $OAZ2_{5Q2}$, $ODC1_{5Q3}$, $SAT1_{5Q1}$, $SMS_{5Q1}$, $SMS_{3Q1}$ and $ARG2_{3Q1}$ (*Figure 2b*). This suggested that activities in the reporter assays possibly originated at the protein level. For $OAZ2_{5Q1}$ reporter activity and mRNA levels were inversely correlated. $ARG2_{5Q1}$, $ODC1_{5Q2}$, $OAZ1_{3Q1}$ and $OAZ3_{3Q1}$, which all suppressed *Renilla* activity (*Figure 2a*), also suppressed the corresponding mRNAs (by 36, 12, 43 and 16%, respectively), consistent with regulation by these four PQS's at least partly at the mRNA level (e.g. transcription). Analysis of the functional PQS sequences suggested seven of the PQS's ($ARG2_{5Q1}$, $OAZ1_{3Q1}$, $ODC1_{5Q2}$, $ODC1_{5Q3}$, $OAZ2_{5Q1}$, $OAZ2_{5Q2}$, $OAZ3_{3Q1}$) likely form quadruplexes with short loops (*Figure 2c*). Their predicted stabilities according to the G-score function QPARS (*Kikin et al., 2006*) (G-score: 16–21) are typical for $G_2$-quadruplexes (*Figure 2c*). The remaining five, which inhibited reporter activity at the protein level, each have one unusually long loop (8–16 nt), and show correspondingly low G-scores (5–13: *Figure 2c*). Three of these $G_2$-PQSs are conserved ($AZIN1_{5Q1}$, $SAT1_{5Q1}$ and $SMS_{5Q1}$) between mouse and human, as well as zebrafish in the case of $SMS_{5Q1}$ (*Figure 2—source data 1*). We performed in vitro translation assays on the five PQS's which comprise one long loop (8–16 nt) and functioned at the post-transcriptional level ($AZIN1_{5Q1}$, $SAT1_{5Q1}$, $SMS_{5Q1}$, $SMS_{3Q1}$ and $ARG2_{3Q1}$).

Reporter mRNAs were in vitro-transcribed and -translated in HeLa lysates (*Figure 2d*, *Figure 2—figure supplement 1*). Compared to matched controls, $SMS_{5Q1}$ and $SMS_{3Q1}$ transcripts produced less *Renilla* activity (28–61%), whereas the $SAT1_{5Q1}$ transcript produced 14-fold more activity, fully consistent with their activities in cells (*Figure 2a,b*). Results from the assays suggested that $AZIN1_{5Q1}$ does not affect translation in this context (*vide infra*). For $ARG2_{3Q1}$ we observed a short transcript which appeared to terminate proximal to the PQS (*Figure 2d*), possibly explaining its unexpected performance during in vitro translation. Indeed, examples of quadruplex structures in 3'UTRs which induce transcription termination have been previously reported (*Kuzmine et al., 2001*; *Wanrooij et al., 2010*).

## Biophysical properties of canonical and long-looped $G_2$-PQS's from PSPs

Multiple experimental techniques are needed to provide conclusive evidence for G-quadruplex formation in vitro (*Lane et al., 2008*). Typically, a PQS is embedded in a short oligoribonucleotide, where it is assumed to fold similarly to its native state independent of flanking sequence. In

comparison to a mutated control, the RNA is then characterized for migration by native polyacrylamide gel electrophoresis (PAGE), by ultraviolet (UV)-melting, by circular dichroism (CD) and by staining with Thioflavin T - the gold standard for monitoring unusual RNA quadruplex structures, which fluoresces upon stacking to RNA G-tetrads and distinguishes quadruplexes from single-stranded- and stem-loop RNAs (*Xu et al., 2016*). We studied four canonical $G_2$-PQS's (ARG2$_{5Q1}$, OAZ2$_{5Q1}$, OAZ2$_{5Q2}$ and ODC1$_{5Q2}$; G scores:~21) and five irregular long-looped $G_2$-PQS's (SMS$_{5Q1}$, SMS$_{3Q1}$, ARG2$_{3Q1}$, SAT1$_{5Q1}$, AZIN1$_{5Q1}$) with much weaker predicted stability (G scores: 5–13). We optimized conditions for the assays using a well-characterized G-quadruplex from the 5'UTR of *NRAS* (NRAS$_{wt}$; G-score: 40) (*Kumari et al., 2007*).

NRAS$_{WT}$ and its mutated control (NRAS$_M$) migrated by PAGE as single bands at different rates (*Figure 3*). Upon staining the gel with Thioflavin T, NRAS$_{WT}$ emitted a higher fluorescence than NRAS$_M$. The UV melt/anneal profile of NRAS$_{wt}$ showed a reversible negative melting transition ($T_M$ 295) typical of intramolecular G-quadruplexes (whereas no transition was observed for NRAS$_M$; *Figure 3—figure supplement 1*). Its CD spectrum displayed a peak at 268 nm and a trough at 239 nm, consistent with a quadruplex or a stem, whereas NRAS$_M$ produced a peak at 273 nm, indicative of unfolded RNA. The four canonical $G_2$-PQS's ARG2$_{5Q1}$, OAZ2$_{5Q1}$, OAZ2$_{5Q2}$ and ODC1$_{5Q2}$ performed similarly to NRAS$_{wt}$ in all of the assays confirming quadruplex structures (*Figure 3*, *Figure 3—figure supplement 1*). Taken together, the data provided strong supporting evidence for control of PA synthesis by $G_2$-PQS's at several points in the PA pathway.

The aforementioned biophysical techniques are less established for irregular quadruplexes, where the structural heterogeneity of long loops and the potential for alternative structures alter typical quadruplex behaviour, for example, shifting peaks to higher wavelengths in CD spectra (*Pandey et al., 2013*; *Bolduc et al., 2016*). Long-looped SMS$_{5Q1}$ and SMS$_{3Q1}$ migrated differently to their controls on PAGE as single bands (*Figure 3* and *Figure 3—figure supplement 2* resp.). Furthermore, Thioflavin T staining produced greater fluorescence emission for SMS$_{5Q1}$ and SMS$_{3Q1}$ RNAs than their controls. A reversible UV melting transition ($\Delta$Abs: ~0.2) with potassium-concentration dependence provided additional evidence of quadruplex formation for SMS$_{5Q1}$ RNA (*Figure 3*) and not for SMS$_{5Q1M}$ (*Figure 3—figure supplement 1*), and SMS$_{5Q1}$ and SMS$_{5Q1M}$ RNAs produced distinct CD spectra with peaks at 278 nm and 273 nm respectively. Together, the data for SMS$_{5Q1}$ provided strong evidence that natural long-looped $G_2$-quadruplexes can form in vitro, and provided a reference profile for such structures. For the weakly-stable SMS$_{3Q1}$ (G-score: 8; 14-nt loop), the CD spectra, the gel migration and the Thioflavin T staining obtained from SMS$_{3Q1}$ and SMS$_{3Q1M}$ confirmed distinct properties for these two short synthetic oligoribonucleotides (*Figure 3—figure supplement 2*) but no UV melting transition was observed, possibly due to limitations of this technique for observing transitions, as proposed previously for similar model structures (*Pandey et al., 2013*).

ARG2$_{3Q1}$ produced multiple bands consistent with multimeric structures on PAGE, which migrated more slowly than ARG2$_{3Q1M}$. Nevertheless, staining with Thioflavin T produced a higher fluorescence for ARG2$_{3Q1}$ and its UV melting profile showed a melting transition typical for a G-quadruplex (with hysteresis) (*Figure 3*), whereas no transition was observed for ARG2$_{3Q1M}$ (*Figure 3—figure supplement 1*). SAT1$_{5Q1}$ RNA also migrated more slowly by PAGE than its control. Although it fluoresced upon Thioflavin T staining, no negative transitions were observed upon UV melting. Its CD spectrum displayed a peak at 264 nm and a trough at 237 nm, whereas SAT1$_{5Q1M}$ was unfolded (*Figure 3—figure supplement 1*). From this data it may be assumed that SAT1$_{5Q1}$ and ARG2$_{3Q1}$, with their two loops and GGGG tracts embedded in these small oligoribonucleotides, are possibly forming intermolecular (quadruplex) structures.

Finally, AZIN1$_{5Q1}$ migrated faster than its control on PAGE, and its CD spectrum indicated base pairing (peak: 268 nm; trough: 233 nm), whereas its control was unfolded. However, AZIN1$_{5Q1}$ produced no fluorescence on Thioflavin T staining and no clear transition during UV melting at 295 nm (*Figure 3*, *Figure 3—figure supplement 2*), casting doubt on a $G_2$-quadruplex structure. On the other hand, UV melting at 260 nm produced a $T_M$ of 49°C (*Figure 3*), suggesting that AZIN1$_{5Q1}$ RNA formed a structure with Watson–Crick base-pairs. This was supported by a mFOLD analysis of the AZIN1$_{5Q1}$-containing sequence, which yielded a stable stem-loop structure (*Figure 3*).

In summary, a variety of measurements added strong supporting evidence that short-looped $G_2$-PQS's, and at least one of the long-looped $G_2$-PQS's formed quadruplexes in vitro. Short oligoribonucleotides containing SAT1$_{5Q1}$ and ARG2$_{3Q1}$ possibly formed intermolecular quadruplex structures. For the two other $G_2$-PQS's the data was either inconclusive, possibly reflecting the limitations of in

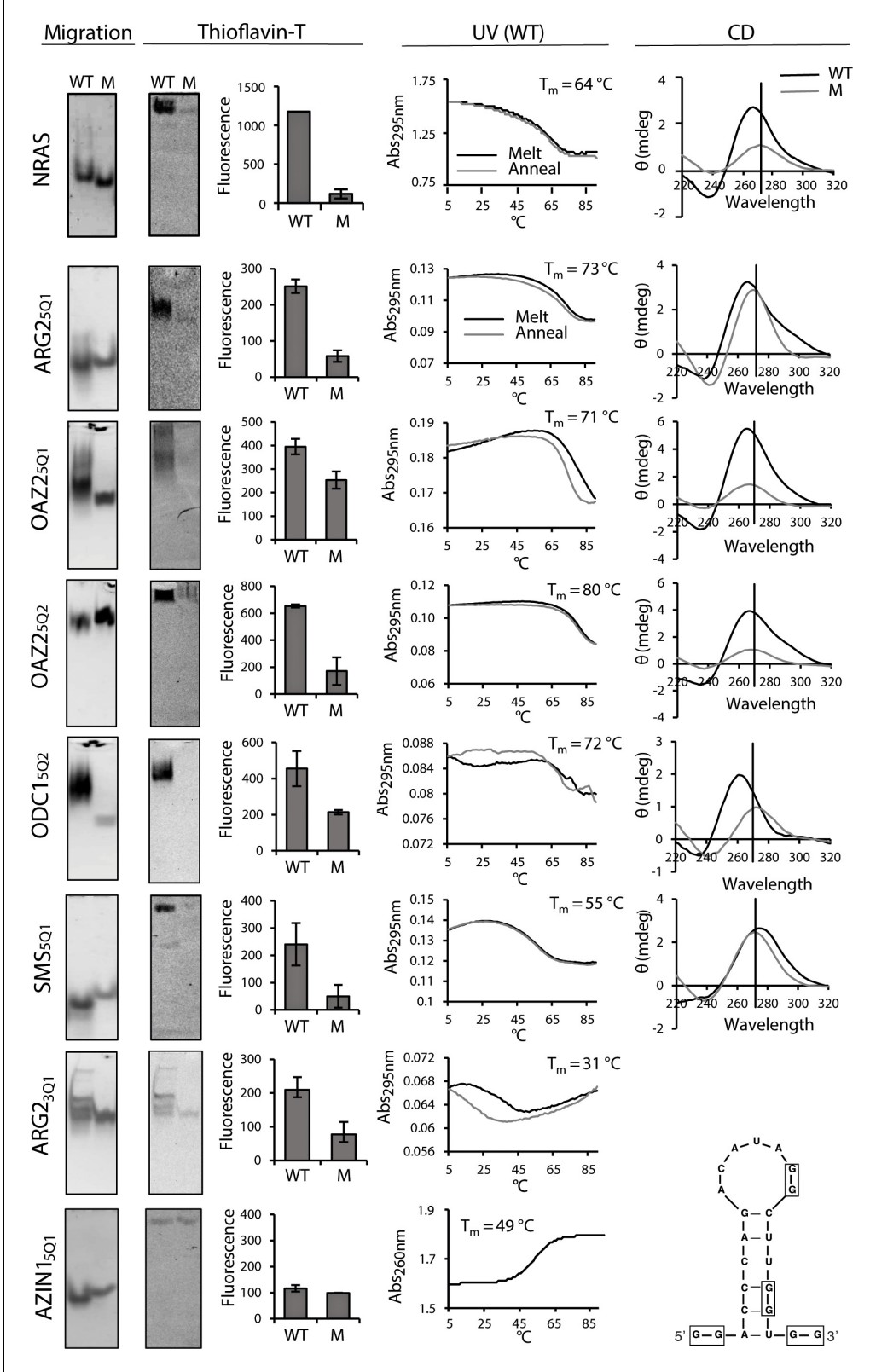

**Figure 3.** Biophysical properties of $G_2$-PQSs from PSPs: Gel migration, Thioflavin T fluorescence, UV (295 nm), CD. All biophysical studies were performed in 100 mM $K^+$. PAGE migration and Thioflavin T staining were performed at different migration times on distinct gels. Melting temps ($T_M$ 295) are shown in the plots; a $K^+$-effect (1 mM and 100 mM $K^+$) on $T_M$ 295 was found for ARG2$_{5Q1}$ (+18.8°C), OAZ2$_{5Q2}$ (+20.9°C) and SMS$_{5Q1}$ (+18.4°C); mutated controls ARG2$_{5Q1M}$, OAZ2$_{5Q1M}$, SMS$_{5Q1M}$, AZIN1$_{5QM1}$, AZIN1$_{5QM2}$, AZIN1$_{5QM3}$, did not show any ($T_M$ 295) melting transitions (data not shown

*Figure 3 continued on next page*

*Figure 3 continued*

and *Figure 3—figure supplement 1*; *Figure 5—figure supplement 3*). CD measurements were not performed for ARG2$_{3Q1}$ due to the presence of multiple RNA species. Error bars represent the standard error (SE) from two independent replicates. (Gels in this figure were cropped: full length gels are in *Figure 3—figure supplement 3* and *4*.

DOI: https://doi.org/10.7554/eLife.36362.006

The following figure supplements are available for figure 3:

**Figure supplement 1.** UV melt-anneal profiles for mutant G$_2$-PQS's.

DOI: https://doi.org/10.7554/eLife.36362.007

**Figure supplement 2.** Additional biophysical data.

DOI: https://doi.org/10.7554/eLife.36362.008

**Figure supplement 3.** Full size migration gels from cropped images.

DOI: https://doi.org/10.7554/eLife.36362.009

**Figure supplement 4.** Full size thioflavin gels from cropped images.

DOI: https://doi.org/10.7554/eLife.36362.010

vitro methods (i.e. SMS$_{3Q1}$) or other competing structures dominated in vitro possibly due to the absence of auxiliary factors present in cells (i.e. AZIN1$_{5Q1}$). Therefore, we looked for evidence for these long-looped quadruplex structures in cells.

## PSP long-looped G$_2$-PQS's perform as quadruplexes in cells

A common means to probe the functionality of a PQS in cells is with pyridostatin (PDS) (*Bugaut et al., 2010*; *Murat et al., 2014*), a ligand which binds to many RNA quadruplexes. When used at micromolar concentrations in vitro, this ligand usually stabilizes quadruplexes. In order to provide further insight on the importance of RNA quadruplexes on PA regulation, we treated cells with PDS, where stabilization of the aforementioned G$_2$-PQS's would be expected to show suppression of PA levels. Using a standard cell viability assay and monitoring $\beta$-actin levels, we first established that PDS could be used without toxicity in HeLa cells at concentrations up to 128 μM (*Figure 4—figure supplement 1*). We then treated cells with PDS and assayed intracellular PAs using a protocol in which PAs from cell lysates are quantified by analytical HPLC (*Morgan, 1998*). Indeed, we found that low micromolar concentrations of PDS strongly reduced spermidine and spermine levels in cells by 70-80% (*Figure 4a*, *Figure 4—figure supplement 2*). To determine whether these effects were due to changes in levels of PSPs, we assayed endogenous SMS and AZIN1 proteins, both of which are expressed in HeLa cells and for which good antibodies are available. PDS treatment produced a 50% decrease in endogenous SMS protein (*Figure 4b*). Next, to examine whether SMS$_{5Q1}$ and SMS$_{3Q1}$ might have contributed to this outcome, we tested PDS on the SMS$_{5Q1}$ and SMS$_{3Q1}$ reporter genes. These assays provided an additional information: they enabled comparison of the effects of stabilizing (increasing) G$_2$-PQS function (with PDS), with the effects of abolishing G$_2$-PQS function by mutation (*Figure 2a*). Indeed, PDS treatment of cells transfected with both SMS$_{5Q1}$ and SMS$_{3Q1}$ reporters dose-dependently reduced luciferase activity by 20-30% compared to matched controls (*Figure 4b*, *Figure 4—figure supplement 3*), as expected for quadruplexes which suppress gene expression, and fully consistent with the effects of mutating SMS$_{5Q1}$ and SMS$_{3Q1}$ (*Figure 2a*). Taken together, the experiments provided evidence in cells that the suppression of PAs by quadruplex-stabilizing PDS (*Figure 4a*) was consistent with reduced SMS protein, and this may have been at least partly due to the additive action of G$_2$-motifs SMS$_{5Q1}$ and SMS$_{3Q1}$ in its UTRs.

PDS (64 μM) yielded a 100-150% increase in both endogenous AZIN1 protein and the AZIN1$_{5Q1}$ reporter (relative to AZIN1$_{5Q1M}$) (*Figure 4c*, *Figure 4—figure supplement 3*). This regulation under reduced polyamine conditions (*Figure 4a*) was in the opposite direction to that of mutating AZIN1$_{5Q1}$ under normal polyamine conditions (*Figure 2a*). Hence, rather than stabilizing the quadruplex we were possibly observing a strong feedback response to global PA suppression by PDS, whereby cells activated AZIN1 in order to raise PA levels, at least partly via the AZIN1$_{5Q1}$ element.

## PSPs self-regulate through some G$_2$-PQS's

Self-regulation and feedback mechanisms are strong features at multiple points in the PA biosynthesis (*Miller-Fleming et al., 2015*; *Ivanov et al., 2010*; *Perez-Leal and Merali, 2012*). We therefore

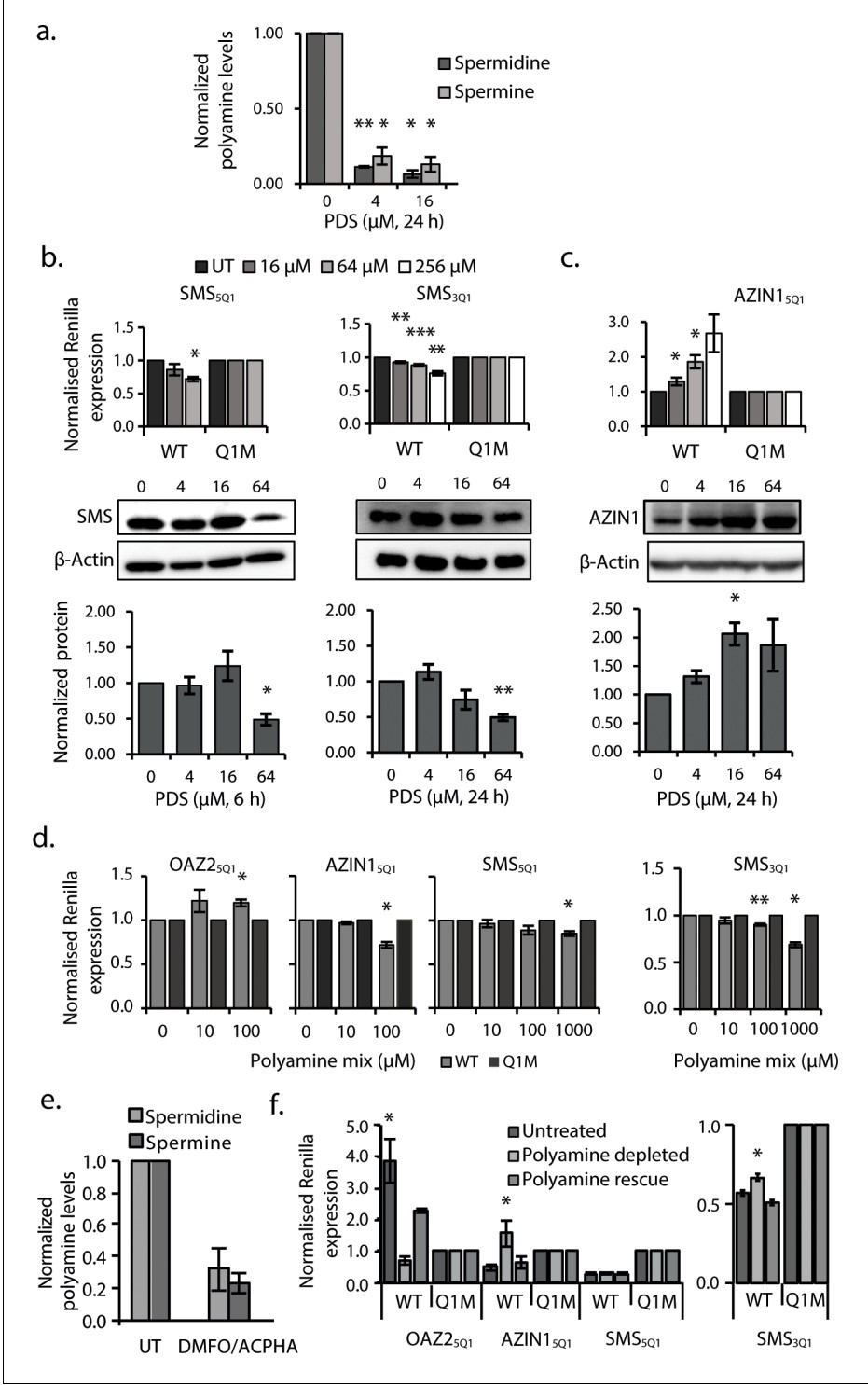

**Figure 4.** $G_2$-PQS in PSPs form quadruplexes in Hela cells and self-regulate. (a) Effects of pyridostatin (PDS) on spermine and spermidine (n=3, *p≤0.05, **p≤0.01). (b) Effects of PDS on endogenous SMS protein at 6 and 24 h (n=3, *p≤0.05, **p≤0.01) (full length blots are in *Figure 4—figure supplement 4*); effects on SMS_{5Q1} (n=3, *p≤0.05) and SMS_{3Q1} (n=3-5, **p≤0.01, ***p≤0.001) wild-type and mutant reporter gene expression (UT: untreated); (c) Effects of PDS on endogenous AZIN1 protein at 24 h (n=3, *p≤0.05) (full length blots are in *Figure 4—figure supplement 4*); effects on AZIN1_{5Q1} wild-type and mutant reporter gene expression (n=6, *p≤0.05). (d) Effect of PA supplementation on reporter activity from PQS's in HeLa cells (n=3, *p≤0.05, **p≤0.01). (e) Effect of DMFO and APCHA on levels of spermine and spermidine in HeLa cells. Two independent replicates

*Figure 4 continued on next page*

*Figure 4 continued*

were performed. (**f**) Effect of PQS's from PSPs on reporter activity in HeLa cells under PA depletion (0.5 mM DFMO, 100 µM APCHA, 6 days) followed by PA rescue (0.5 mM DFMO, 100 µM APCHA, 6 days, 100 µM PA, addition day 5). (n=3-5, *p≤0.05). Error bars represent standard error (SE) from at least two independent replicates. See *Figure 4—figure supplements 5*, *6* and *7*.

DOI: https://doi.org/10.7554/eLife.36362.011

The following figure supplements are available for figure 4:

**Figure supplement 1.** Cell viabilities after PDS treatments of HeLa cells.
DOI: https://doi.org/10.7554/eLife.36362.012
**Figure supplement 2.** Effects of PDS treatment upon spermine and spermidine levels in HeLa cells.
DOI: https://doi.org/10.7554/eLife.36362.013
**Figure supplement 3.** Effects of PDS treatment on PQS reporter genes in HeLa cells: un-normalized data.
DOI: https://doi.org/10.7554/eLife.36362.014
**Figure supplement 4.** Effects of PDS treatment on endogenous polyamine synthesis proteins in HeLa cells.
DOI: https://doi.org/10.7554/eLife.36362.015
**Figure supplement 5.** Polyamine addition and depletion reporter assays.
DOI: https://doi.org/10.7554/eLife.36362.016
**Figure supplement 6.** Polyamine addition and depletion reporter assays.
DOI: https://doi.org/10.7554/eLife.36362.017
**Figure supplement 7.** Polyamine depletion and spermine synthase inhibition.
DOI: https://doi.org/10.7554/eLife.36362.018

investigated whether PAs also self-regulate *via* any of the $G_2$-PQS's in PSP mRNAs. It is well established that under physiological conditions, cells resist uptake of PAs *via* their transporter pathways (*Wallace and Keir, 1986*). Nevertheless, we examined the effects of PA addition (a mix of putrescine, spermine and spermidine at 0.01-1 mM concentrations) to cells expressing a selection of $G_2$-PQS reporters (*Figure 4—figure supplements 5* and *6*). We observed small reproducible statistically-significant decreases in Renilla activity from the $AZIN1_{5Q1}$, $SMS_{5Q1}$ and $SMS_{3Q1}$ reporters after normalization to matched controls, while $SAT1_{5Q1}$ and $OAZ2_{5Q1}$ responded with increased luciferase (*Figure 4d*, *Figure 4—figure supplements 5* and *6*). Once again, given that AZIN1 and SMS drive PA synthesis and SAT1 and OAZ2 are suppressors of PAs (*Figure 1a*), this data also suggested that these motifs participate in feedback loops, whereby cells attempt to maintain normal levels of PAs in the presence of high concentrations of exogenous polyamines.

Next, we turned to two ligands which deplete cells of PAs by distinct mechanisms: D,L-α-difluoromethylornithine (DFMO), which inhibits ODC1, and N-(3-aminopropyl)-cyclohexylamine (APCHA), which inhibits SMS. Consistent with expectations, levels of spermine and spermidine were reduced significantly (to 32 and 24%, respectively) in HeLa cells following combined treatment with DFMO (500 µM) and APCHA (100 µM) for 5 days (*Figure 4e*, *Figure 4—figure supplement 7*). Notably, ligand treatment did not change luciferase activity from most reporter plasmids, which was generally very stable after normalization to matched controls (*Figure 4—figure supplements 5* and *6* and data not shown). However, the ligands increased the activity of $AZIN1_{5Q1}$, and $SMS_{3Q1}$ reporters (114 and 16% respectively), and decreased activity of $OAZ2_{5Q1}$ (81%), compared to controls (*Figure 4f*) (we were unable to assess effects on $SAT1_{5Q1}$ because of too low luciferase counts from $SAT1_{5Q1M}$). To confirm that these effects were due to ligand-induced PA suppression, and not unspecific toxicity, we performed a rescue in which the spermine/spermidine mix was added to the drug-treated cells. The PA mix reversed the effects of the ligands on luciferase counts from $AZIN1_{5Q1}$, $SMS_{3Q1}$ and $OAZ2_{5Q1}$ compared to the controls (*Figure 4f*), and it had no effect on reporters which were unaffected by the ligands (*Figure 4—figure supplements 5* and *6*). Taken together, this further provided evidence that PAs self-regulate through a sub-set of canonical and long-looped $G_2$-PQS's in AZIN1, SAT1, SMS and OAZ2. Other PQS's may also perform similar roles.

## Long-looped AZIN1 $G_2$-quadruplex equilibrates with hairpin structures in vitro

The robust response of the conserved $G_2$-PQS $AZIN1_{5Q1}$ in cells to mutation, to changes in polyamine levels and to PDS treatment suggested that it may play a prominent role in regulation of

polyamine levels. Structure predictions by QGRS Mapper and M-Fold suggested that it might adopt both a G-quadruplex and a hairpin structure, which prompted us to investigate it in further detail, since others have described how RNAs can equilibrate between stem loop and G-quadruplex structures in cation-dependent fashion (*Pandey et al., 2015*; *Mirihana Arachchilage et al., 2015*; *Kwok et al., 2016b*; *Olsthoorn, 2014*). The formation of RNA G-quadruplexes is usually stabilized by $K^+$ (*Lane et al., 2008*), whereas RNA hairpins and duplexes are favoured in the presence of $Na^+$ and $Mg^{2+}$ (*Nakano et al., 2007*; *Tan and Chen, 2008*). Hence, we studied the structural behaviour of $AZIN1_{5Q1}$ (herewith denoted as $AZIN1_{wt}$ for the NMR study), using high-resolution NMR spectroscopy, where we recorded the chemical shifts of its imino proton signals in the presence of different cations. Imino signals at 12–14 ppm are characteristic for Watson-Crick base pairs. The imino signals at approximately 10–12 ppm are characteristic for guanine imino protons involved in H-bonds derived from GU base pairs or from Hoogsteen-like interactions with the oxygen containing acceptor groups of another guanine in the context of G-quadruplexes (*Jin et al., 1990*; *Wang et al., 1991a*, *1991b*; *Smith and Feigon, 1992*; *Bahrami et al., 2012*; *Bugaut et al., 2012*).

We recorded 1D $^1H$ NMR spectra of the five short-looped PQS's (NRAS control, $ARG2_{5Q1}$, $OAZ2_{5Q1}$, $OAZ2_{5Q2}$, $ODC1_{5Q2}$) and the long-looped, less stable $SMS_{5Q1}$ for which biophysical data (Thioflavin T staining and $UV_{295}$-melting; *Figure 3*) indicated the presence of quadruplex structures. All of the short-looped PQS's returned spectra with signals around 11 ppm consistent with G-quadruplex structures (*Figure 5—figure supplement 1*), whereas $SMS_{5Q1}$ did not show signals in this region, likely because of the differing conditions needed/used for NMR experiments compared to the alternate assays (i.e. 500 µM versus 4 µM). We then measured $^1H$ NMR spectra of $AZIN1_{wt}$. The number and position (10.5–13.5 ppm) of the imino signals suggested the presence of multiple structures containing both Watson-Crick and Hoogsteen signals (*Figure 5a and b*). When we increased the concentration of KCl (from 100 to 200 mM), we observed a decrease in intensity for the Watson-Crick imino signals (12–13.4 ppm), but not for those in the Hoogsteen region, consistent with a shift to a quadruplex conformation under increased $K^+$ concentrations (*Figure 5c*). We also performed measurements at 100 mM NaCl or 2 mM $MgCl_2$. In both cases, we observed disappearance of the imino peaks located at 10.8–11.2 ppm, whereas the signals at 11.2–13.5 ppm remained, consistent with the loss of this G-quadruplex and concomitant formation of a double-stranded structure (*Figure 5—figure supplement 1*) (*Bugaut et al., 2012*).

In order to provide additional independent evidence that the signals at 10.8–11.2 ppm indeed derived from a quadruplex conformer, we measured spectra for two mutant sequences. We substituted the first $G_2$-tract at the 5′-end of $AZIN1_{wt}$ with adenines to give $AZIN1_{neg}$, thereby preventing formation of an intramolecular G-quadruplex (*Figure 5a*). Indeed, peaks between 10.8–11.2 ppm disappeared in the $^1H$ NMR spectra (*Figure 5b*). In the second mutant $AZIN1_{pos}$ (*Figure 5a*), the 11-nt long loop was exchanged for a single adenine. This was expected to yield a more stable $G_2$-quadruplex with a shorter loop and prevent the formation of a stem (*Figure 5a*) (*Pandey et al., 2013*; *Zhang et al., 2011*). Indeed, the spectra of $AZIN1_{pos}$ yielded imino signals in a narrow window between 10.8–11.4 ppm, and none in the presumed stem region (*Figure 5b*). Furthermore, superimposition of spectra from $AZIN1_{wt}$ and $AZIN1_{pos}$ revealed substantial overlap in the presumed quadruplex-specific region, supporting the presence of the $G_2$-quadruplex conformer at 100 mM KCl. The small shift downfield of the imino peaks for $AZIN1_{pos}$ might have resulted from an altered environment due to the absence of the long loop L1.

To confirm that the peaks in our spectra around 11 ppm with $AZIN1_{wt}$ were indeed from G imino-protons, we in vitro-transcribed an $^{15}N$-labeled version of $AZIN1_{wt}$ and recorded a $^1H$-$^{15}N$ HSQC (*Figure 5—figure supplement 2*). Similarly to what was previously reported for RNA G-quadruplexes (*Nasiri et al., 2016*), we observed that these protons were bound to $^{15}N$ atoms with a chemical shift of about 145 ppm. This chemical shift is characteristic of a guanine N1 atom bearing a proton involved in a G-quadruplex or GU base-pairing. Imino protons of GU base pairs can be assigned by a $^1H$-$^1H$ 2D NOESY, due to strong NOE's between the imino protons of the guanine and the uridine; in addition, U imino protons in the $^1H$-$^{15}N$ HSQC are expected at approximately 160 ppm for N3. Since neither strong imino-imino NOE's nor H3-N3 imino cross-peaks were observed in our spectra (*Figure 5—figure supplement 2*), we concluded that the cross-peaks observed in the $^1H$-$^{15}N$ HSQC of $AZIN1_{wt}$ between 10–11 ppm in the proton dimension originate from a G-quadruplex structure. G-quadruplexes also produce distinct NOE patterns involving the imino, amino and aromatic protons of the guanine nucleotides (*Jin et al., 1992*; *Macaya et al.,*

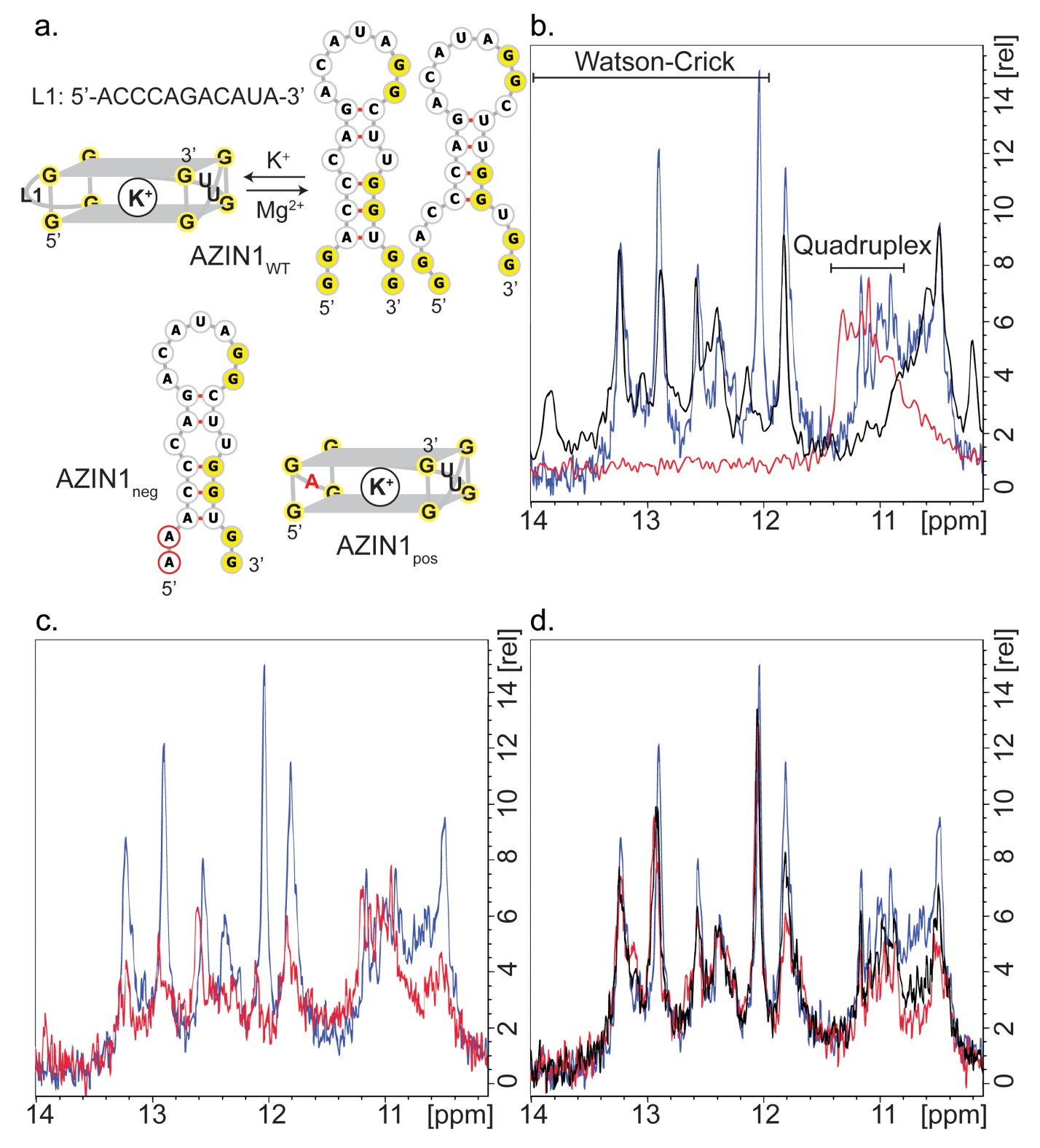

**Figure 5.** Structure of AZIN1wt. (a) Derived model for the equilibrium between two possible hairpin conformers and the $G_2$-quadruplex of AZIN1wt; $K^+$ is expected to favor the $G_2$-quadruplex; $Na^+$ or $Mg^{2+}$ are expected to favor the hairpin conformers, which are predicted by Mfold with free energies $\Delta G$ of $-2.5$ (left) and $-1.8$ kcal/mol (right). The mutants AZIN1neg and AZIN1pos were designed to stabilize the hairpin conformation and the $G_2$-quadruplex, respectively: note the mutation of the first $G_2$-tract at the 5'-end in AZIN1neg and the difference of the $G_2$-quadruplex in the first loop from 5'-end (L1)

*Figure 5 continued on next page*

*Figure 5 continued*

for AZIN1$_{pos}$. Sites of mutation are marked in red. (**b**) Overlay of $^1$H NMR spectra corresponding to the imino region of AZIN1$_{wt}$ (blue), AZIN1$_{neg}$ (black) and AZIN1$_{pos}$ (red) in 100 mM KCl (0.1 mM RNA). (**c**) Overlay of $^1$H NMR spectra corresponding to the imino region of AZIN$_{wt}$ in 100 mM KCl (blue) and 200 mM KCl (red). (**d**) $^1$H NMR spectra after titration of spermine (Spm) to AZIN1$_{wt}$ in 100 mM KCl. Before addition of Spm (blue); after addition of Spm at RNA:Spm ratio of 1:1 (black) and 1:2 (red). Note the stronger decrease of the imino signals < 12 ppm.

DOI: https://doi.org/10.7554/eLife.36362.019

The following figure supplements are available for figure 5:

**Figure supplement 1.** $^1$H NMR spectra of AZIN1$_{wt}$, NRAS, ODC1$_{5Q2}$, ARG2$_{5Q1}$, OAZ2$_{5Q1}$ and OAZ2$_{5Q2}$.

DOI: https://doi.org/10.7554/eLife.36362.020

**Figure supplement 2.** $^1$H $^{15}$N-HSQC and $^1$H $^1$H 2D NOESY of AZIN1$_{wt}$.

DOI: https://doi.org/10.7554/eLife.36362.021

**Figure supplement 3.** Polyamine-related biophysical analysis of AZIN1$_{5Q1}$.

DOI: https://doi.org/10.7554/eLife.36362.022

*1993*; *Smith and Feigon, 1992*). Typical for a G-quadruplex, NOEs between neighbouring imino groups and also between imino and amino groups (shifted to 9–10 ppm compared to Watson-Crick base-pairs) were present in the 2D NOESY spectrum obtained with AZIN1$_{wt}$ (*Figure 5—figure supplement 2*), as reported previously (*Nasiri et al., 2016*).

In conclusion, we accumulated evidence for the coexistence of stem structures and a G-quadruplex structure with the AZIN1$_{wt}$ sequence in 100 mM KCl. In addition, we demonstrated that the nature and concentration of metal ions in solution could direct the equilibrium towards either the stem conformers or the G-quadruplex. To our knowledge this is the first report of a mammalian sequence being in equilibrium between stem structures and a non-canonical RNA G$_2$-quadruplex structure with an exceptionally long loop.

Data from the experiments in cells (*Figure 4*) suggested that PAs might regulate an equilibrium between a stem loop and a G$_2$-quadruplex structure in AZIN1$_{wt}$. Therefore, we also investigated the influence of added spermine on the imino protons of AZIN1$_{wt}$. The proximity of spermine amino groups to the imino protons involved in base pairing might be expected to lower the intensity of their signals through exchange. Indeed, a titration of AZIN1$_{wt}$ with spermine to a ratio of 1:2 revealed that increasing polyamine significantly decreased the imino signals of presumed G-U wobble base pairs (at 11.8 and 10.5–10.8 ppm), as well as the G$_2$-quadruplex signal set (at 10.8–11.2 ppm) (*Figure 5d*) (we were unable to increase the spermine concentration further since at a 1:3 ratio in RNA:Spm precipitation was observed). In addition, we observed small chemical shift perturbations peaks upon spermine addition, which also suggested an interaction. The Watson-Crick imino signals of the hairpin forms (at 12–13.4 ppm) were less affected. Thus, spermine seems to preferentially interact with the quadruplex AZIN$_{5Q1}$ and G-U parts of the stem-containing structures.

Additional supporting evidence for such an interaction of spermine with AZIN$_{wt}$ was available from independent in vitro assays. Incubation of AZIN1$_{5Q1}$ (but not three mutated controls) with spermine in the presence of 1 mM K$^+$ produced a new negative melting transition at 295 nm, consistent with a quadruplex structure ($\Delta$Abs: ~0.02; T$_m$: ~38°C) (*Figure 5—figure supplement 3*). Long-looped SAT1$_{5Q1}$ showed no such change, whereas a minor effect was seen with SMS$_{5Q1}$ (not shown). Under the same conditions, the T$_m$ of the presumed stem measured at 260 nm decreased from 49°C to 28°C (*Figure 5—figure supplement 3*). We also assessed the effects of spermine on AZIN1$_{5Q1}$ and three controls in the Thioflavin T assay (*Figure 5—figure supplement 3*), with the view that if the presence of spermine stabilized formation of a quadruplex, then fluorescence should be increased. Thioflavin T (4.5 μM) was added to pre-annealed AZIN1$_{5Q1}$ and controls after incubation with spermine. Thioflavin T fluorescence was similar for all four sequences in the absence of spermine. However, in the presence of spermine, AZIN1$_{5Q1}$ fluorescence increased 52%. These data provided additional evidence in vitro that spermine might alter the equilibrium dynamics of AZIN1$_{5Q1}$ sub-structures, destabilizing the stem structure in order to form a G$_2$-quadruplex.

## Discussion

Conformational changes in dynamic RNA structures alter gene expression (*Wan et al., 2011*; *Dethoff et al., 2012*). They are triggered by cellular signals such as the binding of RNA helicases,

RBPs, and small ligands, which change the energy landscape of the RNA structure after lowering barriers to conformational exchange (*Dethoff et al., 2012*). The G-quadruplex is one of several important RNA structural elements, that are widespread throughout the transcriptome (*Kwok et al., 2016a*; *Guo and Bartel, 2016*), particularly in UTRs (*Huppert et al., 2008*), where they play roles in transcription termination, polyadenylation, splicing and translation (*Millevoi et al., 2012*). The canonical $G_3$-quadruplex has short-loops and a compact structure that is extremely stable in the presence of cations (*Pandey et al., 2013*). For example, its stability correlates with its capacity to inhibit translation from 5'UTRs (*Halder et al., 2009*) and $G_3$-quadruplexes reportedly need to be unfolded to avoid stalling of reverse transcriptase (*Guo and Bartel, 2016*; *Kwok et al., 2016a*). Using methods that would not identify dynamic (i.e. transiently unfolded) quadruplexes or those strongly folded in the absence of potassium, it was shown that canonical quadruplexes are frequently held unfolded in eukaryotic cells by a dedicated machinery (*Guo and Bartel, 2016*), possibly by specialized helicases (*Bugaut and Balasubramanian, 2012*). However, examples of irregular G-quadruplexes with uncharacterized structures, lower thermodynamic stabilities (in vitro) and distinct biophysical and electronic properties are increasingly proposed. In particular, recent transcriptome-wide searches have identified large numbers of putative G-quadruplexes with $G_2$- instead of $G_3$-tracts, including structures with long loops and even bulged loops (*Kwok et al., 2016a*). $G_2$-quadruplexes are less stable than their canonical $G_3$-counterparts, and long intervening loops further lower their stability in model sequences in vitro (*Pandey et al., 2013*). Indeed, the dynamic character of a metastable G-quadruplex lends itself to participation in equilibria with other secondary or tertiary structures (*Zhang and Balasubramanian, 2012*) where a change in conformation in response to a signal serves to adjust gene expression. Here, we demonstrated using high resolution [1]H NMR spectroscopy one example of such conformational changes under different salt conditions involving the $G_2$-motif present in the 5'UTR of AZIN1.

PAs are regulated by an astounding array of unique and conserved mechanisms involving the enzymes and factors of the PSP pathway (*Lightfoot and Hall, 2014*; *Miller-Fleming et al., 2015*). Many of these mechanisms center on secondary structures and uORFs in the UTRs of PSP mRNAs, thereby affecting translation. Here we describe 12 $G_2$-PQS's, several of which likely form $G_2$-quadruplexes, in the 5'- and 3'UTR's of the PSPs ARG2 (two motifs), ODC1 (two motifs), SMS (two motifs), OAZ2 (two motifs), OAZ3, OAZ1, AZIN1 and SAT1, which were identified using the standard QGRS predictive tool. Seven of them are likely short-looped $G_2$-quadruplexes with typical G-scores. The remainder, which was active at the protein level, all possessed one unusually long loop and returned correspondingly low G-scores. Using matched pairs of wild-type and mutated reporters for each individual $G_2$-PQS, we showed that most of those that affected reporter gene activity would suppress PA synthesis in their native UTR setting (*Figure 2*). We employed multiple assays on nine PSP $G_2$-PQS's in vitro, comparing them to a $G_3$-quadruplex reference (*Figure 3*). The four short-looped $G_2$-PQS's behaved similarly to the reference providing strong evidence for a quadruplex structure. Three of the long-looped $G_2$-PQS's ($SMS_{5Q1}$, $ARG2_{3Q1}$, $AZIN1_{5Q1}$) also showed many characteristics typical of a G-quadruplex in vitro, albeit with lower stabilities. In addition, data from cell assays provided persuasive evidence for the structure and function of long-looped $G_2$-quadruplexes (*Figure 4*).

PDS, a ligand which generally stabilizes RNA G-quadruplexes, altered the expression of reporters containing the $G_2$-PQS's from SMS and AZIN1 UTRs, and also levels of native SMS and AZIN1 proteins in cells. Small effects (20–30%) on the individual reporters appeared at PDS concentrations of 16–256 μM, however 15-fold lower ligand concentrations sufficed to reduce endogenous spermine and spermidine levels by >70–80%. We took this as a possible indication that PDS acts additively on individual $G_2$-quadruplexes in the pathway to produce a strong cooperative suppression of PAs. In addition, several $G_2$-PQS's responded to changing levels of PAs in cells with feedback, for example PA addition suppressed the reporters from SMS and AZIN1 (drivers of PA synthesis), and activated those of SAT1 and OAZ2 (PA inhibitors).

G-quadruplexes in regulatory regions affect gene expression in several different ways depending on their structures and their positions. Indeed, in our study most of the $G_2$-PQS's decreased - but a few increased - expression of their host genes; some affected protein levels exclusively, while others were active on the RNA levels. The majority of the motifs scored positively in assays consistent with a regulatory role in PA biosynthesis. In some cases they may synergize with other regulatory elements, such as uORFs that collectively serve as PA sensors, making minor individual contributions -

and a major collective contribution - to the control of PA levels in an RNA structure-driven network. This is in line with a recent hypothesis that PAs are regulated by a common, yet undiscovered chemical or physical mechanism (*Miller-Fleming et al., 2015*). One possible function of such a network would be to control expression of the 'polyamine modulon' (*Igarashi and Kashiwagi, 2011*), a family of proteins whose translation is regulated by PAs in bacteria, yeast and mammalian systems to control cell proliferation. Misregulation of PAs is also directly associated with tumorigenesis and several polyamine inhibitors have been investigated in cancer clinical trials (*Casero and Marton, 2007*). The response of the PQS's in PSP genes to PAs may help guard against cues which increase proliferation and cellular transformation/migration.

A principle function of PAs in cells is to bind and stabilize RNA structures. The monocations $K^+$, $NH_4^+$ and $Na^+$ strongly stabilize G-quadruplexes by binding in the G-quadruplex cavity (*Guiset Miserachs et al., 2016*). It is therefore conceivable that the charged terminal amines of PAs may bind directly to $G_2$-quadruplexes, possibly substituting for $K^+$ and $Na^+$ in the tetrad. Indeed, polyamines have been reported to stabilize, destabilize and alter DNA quadruplex structures in vitro (*Keniry and Owen, 2014*, *2013*; *Wen and Xie, 2013*; *Kumar et al., 2009*; *Keniry and Owen, 2007*; *Miyoshi et al., 2003*; *Keniry, 2003*; *Qi et al., 2014*) and in one case evidence for polyamine-modulation of a DNA quadruplex in cells was presented (*Kumar et al., 2009*). In our in vitro study, AZIN1$_{5Q1}$ switched between a stem-loop and a G-quadruplex in the presence of different concentrations of salt and spermine. Overall, this behaviour may be reminiscent of the Iron Responsive Element, which is an RNA stem loop in the 5'UTR of mRNAs that encodes a family of proteins required for iron metabolism and homeostasis (*Theil, 2015*); binding of $Fe^{2+}$ to the IRE alters translation of these mRNAs as a coordinated feedback control mechanism. Our work may indicate a similar role for PAs in PA self-regulation though altering or switching RNA structural elements present within PA biosynthesis genes.

## Materials and methods

### Cell culture

HeLa cells (ATCC, CCL-2, STR profiling and Mycoplasma tested) were maintained in DMEM (Gibco) supplemented with dialyzed fetal bovine serum (10%, Invitrogen) and aminoguanidine (1 mM, TCI Chemicals) in humidified 5% $CO_2$ atmosphere at 37°C. To deplete cellular polyamines, $1.65 \times 10^5$ cells were grown in a T75 flask in the presence of DFMO (Bachem, 0.5 mM, DMEM) and APCHA (ABCR, 0.100 mM, DMEM) for 6 d (dosing at d0 and d2). To analyze the effect of polyamine supplementation during polyamine depletion, polyamines (Sigma Aldrich, 0.1 mM, DMEM) were added to the drug-treated cells on d5. Cells did not appear to be unduly stressed by the treatments, though proliferation was clearly slowed (cell confluency at d5: 60-70% untreated; 40-50% drug treated). The quadruplex stabilizing ligands and PDS (Sigma Aldrich) and the polyamine mix (ratio 1:1:1 putrescine, spermidine and spermine) were dissolved fresh in DMEM and used at the respective concentrations. PDS and polyamine mix were added to cells directly prior to plating for the reporter assays. Cell cytotoxicity of treatments was measured using a standard assay: it was not observed for PDS concentrations ≤128 μM.

### Bioinformatics

Sequences of the 3'-UTRs and 5'-UTRs of the PSPs were obtained from Ensembl (http://www.ensembl.org/index.html) (*Supplementary file 1*). The PSP UTRs selected were upstream of known protein coding sequences chosen based on their consistent annotation by multiple public resources (CCDS project and RefSeq) (*Supplementary file 1*). If several known transcripts fitting this criterion were available, an example was selected which contained quadruplexes conserved across ≥1 transcripts (*Supplementary file 1*). PQS were identified using the QGRS mapper (*Kikin et al., 2006*) and quadruplex phylogenetic conservation was assessed using QGRS-H mapper (*Menendez et al., 2012*) and through clustral-W alignment.

### Reporter plasmid construction

Selected PSP UTR sequences where amplified from either synthetic oligonucleotides, cDNA encoded plasmids or cDNA reverse-transcribed from Huh7 or HeLa cells, using the M-MLV reverse

transcriptase (Promega) according to the manufacturer's instructions. Amplified sequences were sub-cloned into a dual luciferase psi-check-2 vector (Promega) upstream of the *Renilla* luciferase gene in the NheI single cloning site or downstream of the *Renilla* luciferase gene in the Not1-Xho1 multiple cloning site. Correct insertion was confirmed by DNA sequencing. For the 5' UTRs, a modified psi-check vector placing the start codon at the natural position directly adjacent to the 5'-UTR insert was used (*Calvo et al., 2009*). All primers, cDNA plasmids and synthetic oligonucleotides used for reporter plasmid construction are listed in *Supplementary file 1*.

## Site directed mutagenesis

Site directed mutagenesis was performed using the Q5 Site-Directed Mutagenesis Kit (NEB) with primers listed in *Supplementary file 1*. Mutagenesis was confirmed by DNA sequencing. Due to technical reasons four quadruplexes present in and at the 3' of a predicted uORF in the 5'-UTR of ODC1 and one quadruplex present in its 3'-UTR were excluded from this study. In addition, due to the complex nature of the OAZ2 3'-UTR with its 11 predicted quadruplexes, this was also excluded from this study.

## Reporter assay transfection

Cells (6 x $10^3$ /well) were seeded in white 96-well plates. Plasmid DNA (20 and 40 ng for 3' and 5' UTR reporter plasmids, respectively) was transfected using JetPEI (Polyplus) according to the manufacturer's instructions.

## Reporter protein analysis

Cells were lysed and both Firefly and *Renilla* luciferase activities were measured at 36 h post-transfection using the Dual-Glo Luciferase Assay System (Promega), according to the manufacturer's instructions. *Renilla* luciferase activity levels were normalized to that of firefly luciferase activity. Wild type *Renilla* expression levels were normalized to that of the quadruplex destabilizing mutant.

## Intracellular polyamine extraction and HPLC quantification

Intracellular polyamine extraction was carried out according to the procedure described by Morgan (*Morgan, 1998*). Cell pellets were homogenized by the addition of 50% TCA (2%) in PBS (100 μL/ $10^6$ cells), followed by vigorous mixing. A synthetic amine, 1, 8-diaminooctane (Sigma Aldrich) was added to the crude mixture as an internal standard. The precipitated protein and cell debris were sedimented by centrifugation (10 min, 17,000 rpm, RT). The supernatant was removed, and both pellet and supernatant stored at -20°C until required. Before benzoylation, NaOH (4 eq., 2N) was added to the supernatant and the solution mixed vigorously. Following addition of benzoyl chloride (0.4, 50% solution in methanol) the resulting mixtures were incubated at RT for 45 min (with intermittent vortexing at 5 min intervals). Polyamines were extracted with chloroform (1 eq.), washed with $H_2O$ (0.5 eq.) and evaporated to dryness. The residue was re-suspended in acetonitrile (42%), filtered and analyzed by HPLC. Benzoylated samples were injected on to an Agilent Eclipse XDB-C18 (250 mm) column and using an isocratic solvent system (42% acetonitrile) at a flow rate of 0.5 mL/min, benzoylated spermidine, spermine and, 1, 8-diaminooctane were detected at 198, 224 and 254 nm at 3.0, 5.2 and 5.6 min respectively. Polyamine levels of treated were normalized to that of untreated samples. The cellular polyamine levels were normalized to that of the amount of protein in the pellet. Pellet protein levels were quantified through the BCA protein assay (Thermo Scientific).

## Reporter RNA analysis

Total cellular RNA was purified by the RNeasy mini kit with on-column DNase digestion. To ensure contaminating plasmid would not hinder mRNA analysis, DNase digestion (on-column) was performed five times and the appropriate controls (without reverse transcriptase) were utilized in the RT-qPCR reactions. Total cellular RNA (0.5 μg) was reverse transcribed by M-MLV reverse transcriptase according to the manufactures instructions. mRNA expression levels were quantified using Fast-Start essential DNA green master (Roche) in a LightCycler 480 (Roche), again according to the manufactures instructions. All samples were measured in triplicates. The mRNA levels were normalized to that of the co-transfected Firefly gene and the fold changes calculated using the ddCt algorithm. Sequences for primers utilized are:

*Renilla*: F: AAGAGCGAAGAGGGCGAGAA; R: TGCGGACAATCTGGACGA;
Firefly: F: CGACTTCGTGCCAGAGTCTT; R: GTACATCAGCACCACCCGAA

## In vitro transcription

Plasmids were linearized using the PmeI (AZIN1) or NOTI restriction enzyme, which cuts at the 3′ end of the coding region of the luciferase reporter gene after the 3′-UTR. 5′-capped transcripts were generated in vitro using the mMESSAGE mMACHINE T7 kit (Ambion), following the manufacturer's instructions. The RNA concentration was determined by UV spectroscopy. The integrity and the size of each transcript were confirmed by 2% agarose gel analysis.

## In vitro translation

In vitro translation of 1500 ng of in vitro-transcribed mRNAs was carried out in a cell-free translation system consisting of extracts from nuclease-treated HeLa lysate (ThermoScientific, #88881).

## Western blot

For protein analysis, cells were washed twice with ice cold PBS and resuspended in lysis buffer for protein extraction (1x PDS, 1% Triton X-100, protease inhibitor cocktail (Roche)). Cell lysates were obtained by centrifugation at 14.000 x g for 10 min at 4°C. Protein concentrations were estimated by the Bio-Rad protein assay using bovine serum albumin as standard. Lysates were separated by SDS/PAGE on a mini-PROTEAN TGX gel (BIO-RAD). Proteins were subsequently transferred to a polyvinylidene difluoride transfer membrane (Roche) using a transfer semidry blot cell (BioRad Laboratories), blocked (5% milk in 1x PBS-T, 50 mins, RT) and incubated with the appropriate antibody: anti-human SMS (Origene, TA503099, 1/1000) and anti-human AZIN1 (Sigma-Aldrich, WH0051582M1, 1/1000) in blocking buffer, overnight at 4°C. Immunoblots were visualized with ECL Prime Western Blotting Detection Reagent (GE Healthcare) using horseradish peroxidase-labelled secondary antibodies (1 h; RT; blocking buffer). To confirm equal loading in each sample, the membranes were stripped in stripping buffer (200 mM Glycine, 3.5 mM SDS and 0.0001% Tween20) and re-blotted with anti-β-actin (Santa Cruz Biotechnology, sc-69879). The images were captured and analysed with a ChemiDoc Touch Imaging System (BIO-RAD). Densitometric analysis was performed using Image Lab software (BIO-RAD).

## RNA preparation and annealing

RNA samples were prepared as described in reference (*Guennewig et al., 2014*). Prior to their use, RNA was prepared in annealing buffer containing lithium cacodylate buffer (10 mM. pH 7.2), KCl (100, 10 or 1 mM) with and without spermine or spermidine (at the stated concentrations). RNA samples were heated at 90°C for 10 min and immediately slow cooled to 5°C at a controlled rate of 0.2°C min$^{-1}$.

## Gel migration assay

Oligoribonucleotide concentrations were monitored by OD and adjusted to ensure equal loading across sample wells. Glycerol (50%) was added to pre-annealed oligonucleotides (see RNA preparation and annealing) at a final concentration of 10% prior to gel loading on to a 15% native PAGE gel. Following sample loading, the gel was run at 50 V for 2-3 h. Gels were stained with 1× SYBR Gold (Life Technologies) for 40 min before visualization on a Gel Doc XR (BioRad). Differences in Syber gold staining of the WT and M oligonucleotides is likely due to the sequence-selective binding nature displayed by Syber gold (*Tuma et al., 1999*).

## Thioflavin T assay

Oligoribonucleotides were prepared and gel-loaded as above. Following sample loading, the gel was run at 50 V for 2-3 h. Gels were incubated in 0.5 μM 3,6-dimethyl-2-(4-dimethylaminophenyl) benzothiazolium (ThT, Sigma-Aldrich, tris-borate-ethylenediaminetetraacetic acid) for 15 min under gentle agitation and briefly washed in water before visualization on a Typhoon Trio+ Imager (GE Healthcare). ThT stained bands were quantified using imageQuant™ software and used to calculate the average amount of signal detected.

## UV melting

UV melting experiments were performed using a Varian Cary 300 spectrophotometer. RNA was prepared as described in RNA preparation and annealing above. The UV absorbance profiles were recorded at 295 and 260 nm during one cycle of cooling/heating between 90 and 5°C at a rate of 0.2°C/min. Thermodynamic parameters and melting temperature ($T_m$) values were derived as described in reference (*Mergny and Lacroix, 2009*).

## Circular dichroism

CD experiments were performed using a Jasco J-10 spectropolarimeter. RNA was prepared as described in RNA preparation and annealing above. Two CD scans over the wavelength range of 220 to 320 nm were performed at 50 nm min$^{-1}$ with a 2 s response time, 1 nm pitch and 1 nm bandwidth, and the average taken. For each experiment, a CD spectrum of buffer alone was recorded and subtracted from the spectrum obtained for the oligonucleotide containing solution.

## ThT fluorescence measurements (plate reader)

Thioflavin T staining is the gold standard for identification of novel quadruplex structures (*Xu et al., 2016*). The experiments were carried out in 96-well microplates. The annealed RNA samples were incubated with spermine for 1 hr at RT. ThT (4.5 µM) was added and measurements were performed at room temperature. The fluorescence emission was collected at 487 nm with excitation at 440 nm in a microplate reader (Tecan Spark 20M).

## NMR spectroscopy

RNA was prepared to a concentration of 0.03-0.5 mM in 10 mM lithium cacodylate, 1 mM KCl or LiCl, pH 5.8 using 3'000 NMWL Amicon Ultra Centrifugal Filters (Merck Millipore Ltd., IRL). After heating to 95°C for 5 min KCl (100 or 200 mM), NaCl (100 mM) or MgCl$_2$ (2 mM) was added directly to the warm solution containing the oligonucleotides. The samples were then progressively cooled down to room temperature for 60 min before to be stored on ice. One-dimensional watergate $^1$H NMR spectra 2D $^1$H $^{15}$N HSQC and 2D $^1$H-1H noesy spectra were recorded at 288 K on Bruker AVIII-500, 600 and 900 MHz spectrometers equipped with a cryoprobe. Topspin 2.1 (Bruker) was used for data processing. The secondary structure of the hairpin RNA was illustrated with the RNA visualization toolforna (*Kerpedjiev et al., 2015*).

## In vitro transcription

$^{15}$N labelled RNA was produced by in vitro transcription from double-stranded DNA templates (Microsynth AG, Balgach, Switzerland) using T7 polymerase and was subsequently purified by denaturing HPLC followed by butanol extraction as previously described (*Duss et al., 2010*). In order to increase the yield for AZIN1$_{wt}$ a modified DNA template was used where the two 5'-terminal nucleotides of the template strand were 2'-O-methyl-modified (*Kao et al., 2001*)

## Independent replicates/biological replicates and biological significance

independent replicates represent experiments performed with independent batches of cells and reagents. Data are represented as the mean ± standard error (SE). P-values were calculated for cell or lysate based experiments with ≥3 biological replicates using the t-test.

## Acknowledgements

We thank A Camus, M Jayakumar, D Kayalar, F Schmitz-Hübsch, Q Nguyen, T Hua, L Isenman, L Thommen and A Laski for help with cloning and/or preliminary experiments, and N Luedtke for comments on the manuscript. This work was supported in part by grants from the ETH (to JH), the NCCR RNA and Disease, funded by the Swiss National Science Foundation (to FHTA) and the Novartis Foundation (formerly the Ciba Geigy Jubilee Foundation) (to HLL).

## Additional information

### Funding

| Funder | Grant reference number | Author |
|---|---|---|
| Novartis Foundation | | Helen Louise Lightfoot |
| Schweizerischer Nationalfonds zur Förderung der Wissenschaftlichen Forschung | NCCR RNA and Disease | Antoine Cléry<br>Frédéric Hai-Trieu Allain |

The funders had no role in study design, data collection and interpretation, or the decision to submit the work for publication.

### Author contributions

Helen Louise Lightfoot, Conceptualization, Formal analysis, Supervision, Funding acquisition, Writing—original draft, Project administration, Writing—review and editing; Timo Hagen, Conceptualization, Resources, Data curation, Formal analysis, Supervision, Validation, Investigation, Methodology, Writing—original draft, Writing—review and editing; Antoine Cléry, Conceptualization, Resources, Formal analysis, Validation, Investigation, Visualization, Methodology; Frédéric Hai-Trieu Allain, Conceptualization, Resources, Formal analysis, Validation, Investigation, Methodology, Writing—review and editing; Jonathan Hall, Conceptualization, Resources, Formal analysis, Supervision, Project administration, Writing—review and editing

### Author ORCIDs

Helen Louise Lightfoot (iD) http://orcid.org/0000-0002-9292-0087
Timo Hagen (iD) http://orcid.org/0000-0003-0139-6011
Antoine Cléry (iD) http://orcid.org/0000-0002-4550-6908
Frédéric Hai-Trieu Allain (iD) http://orcid.org/0000-0002-2131-6237
Jonathan Hall (iD) http://orcid.org/0000-0003-4160-7135

### Decision letter and Author response

Decision letter https://doi.org/10.7554/eLife.36362.026
Author response https://doi.org/10.7554/eLife.36362.027

## Additional files

### Supplementary files

• Supplementary file 1.
DOI: https://doi.org/10.7554/eLife.36362.023

• Supplementary file 2.
DOI: https://doi.org/10.7554/eLife.36362.024

### Data availability

All data generated or analysed during this study are included in the manuscript and supporting files.

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
