## [Decision Letter]

[Editors’ note: a previous version of this study was rejected after peer review, but the authors submitted for reconsideration. The first decision letter after peer review is shown below.]

Thank you for submitting your work entitled "Control of the polyamine biosynthesis pathway by RNA G_2_-quadruplexes" for consideration by *eLife*. Your article has been reviewed by three peer reviewers, and the evaluation has been overseen by Reviewing Editor Douglas Black and Senior Editor James Manley. The following individuals involved in review of your submission have agreed to reveal their identity: Cyril Dominguez (Reviewer #2).

Our decision has been reached after consultation between the reviewers. Based on these discussions and the individual reviews below, we regret to inform you that your work cannot be considered further for publication in *eLife*.

There was general agreement among the reviewers and the reviewing editor that the identification of regulatory G-quadruplexes was of some interest and that several findings had merit. However, it was also thought that the analysis did not go very deep. Instead of doing a limited analysis of multiple elements, it would be more significant to do a conclusive analysis of one or two. Notably, neither the existence of the identified elements as quadruplexes, especially those with 0-length loops, nor the role of the elements in controlling polyamine biosynthesis, were demonstrated to a high degree of certainty. For the study to be of interest to the broad audience of *eLife*, these findings would need much stronger support. Generating such data would take much longer than the maximum 8 weeks allowed for a revision. Thus, although some of the initial responses were positive, the consensus decision was to decline the manuscript.

Reviewer #1:

In this paper Hall and Lightfoot provide a survey of locations of possible G quartet structures within 5' and 3' UTRs of mRNAs for enzymes in the polyamine pathway(s) of mammalian cells. Their interesting idea is that perhaps since polyamines might interact with G-quartets, then G-quartet structures might signal the presence of regulatory elements that respond to changes in polyamine levels in the cell.

They develop several assays for their analysis – expression of the target gene as wt and as a mutation designed to ablate the putative G-quartet (PQS) in vitro and in cells – comprise one set of assays. Other assays include structural and dye binding assays on model RNAs that are designed to distinguish G-quartets from other RNA folds. Finally, a bold in vivo approach is taken to hyperstabilize G-quartets with pyridostatin in vivo and see how this affects expression of endogenous genes and reporters, and the results are mostly consistent with the mutational data.

A major concern is that there are too many genes studied at too shallow a depth. There are many lower resolution and correlative assays employed so that the conclusions rely on the aggregate acceptance of many different assays each with different alternative explanations, and some of which work better on some genes and selected elements than others. The assays for structure determination aren't nucleotide resolution, and it remains tricky to predict these structures confidently by computational methods. Overall the need to accept this as an aggregate survey weakens the conclusiveness of the presentation. Having done this interesting survey, the authors are likely interested in higher resolution study of one or a few of these elements in detail that would really nail down the idea that G-quartets interact with polyamines to control expression of polyamine metabolic enzymes.

Reviewer #2:

In the manuscript, Lightfoot and Hall describe the effect of transient G_2_-quadruplexes on the expression levels of proteins involved in the polyamine biosynthesis pathway.

Enzymes involved in this pathway are regulated at different levels including transcriptional and post-transcriptional levels.

The authors hypothesized that these levels could be regulated at the post-transcriptional level by specific RNA structures, G-quadruplexes. Indeed, RNA G-quadruplexes have already been shown to play a role in RNA metabolism regulation, such as translation, splicing,.

Using bioinformatic tools, the authors identified putative G-quadruplex forming sequences (PQS) in the 5' and 3'UTR of many genes involved in this pathway. They validated the effect of these putative G-quadruplexes using in vivo and in vitro assays on WT and mutated sequences, as well as the effect of a well-characterized G-quadruplex stabilizer.

Overall, the manuscript is well written, clear, and sound. The authors nicely demonstrate that while many PQS do not have any effect on protein expression, other have a significant influence. The study of mutated sequences in itself is controversial (is the effect of the mutation due to a disruption of the G-quadruplex or a change in the ratio of proteins bound is always difficult to assess) but the complementary approach of testing the effect of a G-quadruplex stabilizer on protein expression makes the conclusions strong and compelling.

The final conclusion of the manuscript that spermine itself stabilizes G-quadruplex is interesting but lacks in my view strong evidence. The in cells experiments cannot provide evidence that the observed effects are direct and the in vitro validation assays have been done only on one PQS (AZIN1). It would be interesting to test the effect of spermine of the stability of other PQS sequence that respond to spermine in cells such as SMS_5Q1_, SMS_3Q1_, SAT1_5Q1_ and OAZ2_5Q1_. It would also be interesting to compare the effect of spermine and PDS on G-quadruplex stabilization in vitro.

Subsection “PSP long-looped G_2_-PQSs perform as quadruplexes in cells”: The authors observed that the effect of PDS on AZIN1_5Q1_ was opposite to what was expected based on the mutant results. However, AZIN1 also has a PQS in its 3'UTR (AZIN1_3Q1_). It would be interesting to test the effect of PDS on this reporter.

Reviewer #3:

This is a well described study by Lightfoot and Hall that demonstrates many G-rich sequences regulate the RNA and protein levels of proteins important in polyamide (PA) biosynthesis. The authors identify many putative G-quadruplexes in the 5' and 3' UTRs of PA protein-coding genes and show that some of these elements can form G-quadruplex structures in vitro and provide evidence that they may function via these structures using a small molecule known to bind G-quadruplex structures. This work adds to our understanding of how these genes are regulated through potential G-quadruplex structures. Although this is an interesting body of work, several concerns reduce enthusiasm for publication in *eLife*.

1) A major concern is that there doesn't appear to be any data in this manuscript or other publications (search by this reviewer of the literature) that show G-quadruplexes can form without a single nucleotide loop (usually defined as 1-7 nucleotides although longer loops exist as previously demonstrated and in this work). If these references exist it would be helpful for the authors to include. Structurally, it doesn't seem possible for the four Gs to be split into two parts of the G-quad. This should be demonstrated or explained if previously published data hasn't demonstrated this unique G-quadruplex structure before. This point applies to the elements in SAT1, AZIN1 and OAZ3.

2) Presumably the authors tested if polyamines bind and stabilize / destabilize the putative G_2_ quadruplex structures. These results should be included or discussed because otherwise the reader is left wondering how the polyamines self-regulate except for the AZIN1 example. For the AZIN1 data, the authors should correct ΔAbs of ~0.2 to ~0.02 in the Results section (Long-looped AZIN1 G_2_-quadruplex is induced by PAs in vitro). Again, please provide a rationale for how a G-quad can form without a loop nucleotide (see point #1) and not just state that high resolution structural studies will need to be performed (Discussion section). If the authors demonstrate G-quads can form without a loop this would be a major finding for the field and would be of great interest. The data for the 3 putative G-quads lacking loops is weak compared to the data provided for the G-quads with loops.

3) A more minor concern is the data shown in Figure 2. Several of the gel images for the Thioflavin-T with the G-quadruplexes don't appear to match the quantitation shown in the next panel although this could be explained by loading differences. For example, for ODC1_5Q2_ there is a 2-fold difference in fluorescence quantitation, which doesn't seem to match the adjacent gel image. This might be explained by the significant difference in the loading of RNA seen in the first panel. If the RNA concentration levels or ability to detect RNA is significantly different for some species it may call into question the ability to detect differences between WT and mutant RNAs.

4) A minor point is why do the authors not spend more time in the discussion on the most significant PA regulatory element they identified in SAT1 (30-fold change)?

[Editors’ note: what now follows is the decision letter after the authors submitted for further consideration.]

Thank you for resubmitting your article "Control of the polyamine biosynthesis pathway by G_2_-quadruplexes" for consideration by *eLife*. The evaluation has been overseen by Douglas Black as Reviewing Editor and James Manley as the Senior Editor. Your revised article has been reviewed by two of the previous reviewers and one new reviewer. The following individual involved in review of your submission has agreed to reveal his identity: Cyril Dominguez.

The reviewers have discussed the reviews with one another and the Reviewing Editor has drafted this decision to help you prepare a revised submission. It was generally agreed that the submission was much improved, but additional work was needed.

Essential revisions:

Given the very loose restraints on the G_2_ search criteria it is not surprising that many sequences were found. However, it seems possible that the G_2_s could be involved in alternative secondary structures that are not quadruplexes. Have the authors checked the PQS's for other secondary structures (for example, by M-fold?).

Transfected dual luciferase assays can be highly variable and the gold standard would be to report errors from at least 3 independent replicates, not 2. For example, AZIN1 has a strong effect in the dual luciferase assay (Figure 2A) but does not behave like a quadruplex in the biophysical assays in Figure 3. This could be an indication that the dual luciferase assay is not necessarily reporting on quadruplex formation. Negative controls are needed showing that mutations in the predicted loop regions have no effect in the dual luciferase assay.

AZIN1 showed no evidence for quadruplex formation in the biophysical assays (Figure 3), yet this is the only RNA that was chosen for further characterization by NMR in order to demonstrate quadruplex formation. The rationale for pursuing AZIN1 needs to be explained.

How does the addition of polyamines affect the biophysical characterizations in Figure 3?

Figure 5. The ^1^H signals around 11 ppm are consistent with quadruplex formation but this is not conclusive as other types of base pairs can also give rise to resonances in this region of the proton spectrum. A ^1^H-^15^N through hydrogen bond NMR experiment is needed to really show this. Do the RNAs that respond to Thioflavin T also have similar 1D NMR spectra?

---

## [Author Response]

[Editors’ note: the author responses to the first round of peer review follow.]

There was general agreement among the reviewers and the reviewing editor that the identification of regulatory G-quadruplexes was of some interest and that several findings had merit. However, it was also thought that the analysis did not go very deep. Instead of doing a limited analysis of multiple elements, it would be more significant to do a conclusive analysis of one or two. Notably, neither the existence of the identified elements as quadruplexes, especially those with 0-length loops, nor the role of the elements in controlling polyamine biosynthesis, were demonstrated to a high degree of certainty. For the study to be of interest to the broad audience of eLife, these findings would need much stronger support. Generating such data would take much longer than the maximum 8 weeks allowed for a revision. Thus, although some of the initial responses were positive, the consensus decision was to decline the manuscript.Reviewer #1:In this paper Hall and Lightfoot provide a survey of locations of possible G quartet structures within 5' and 3' UTRs of mRNAs for enzymes in the polyamine pathway(s) of mammalian cells. Their interesting idea is that perhaps since polyamines might interact with G-quartets, then G-quartet structures might signal the presence of regulatory elements that respond to changes in polyamine levels in the cell.They develop several assays for their analysis – expression of the target gene as wt and as a mutation designed to ablate the putative G-quartet (PQS) in vitro and in cells – comprise one set of assays. Other assays include structural and dye binding assays on model RNAs that are designed to distinguish G-quartets from other RNA folds. Finally, a bold in vivo approach is taken to hyperstabilize G-quartets with pyridostatin in vivo and see how this affects expression of endogenous genes and reporters, and the results are mostly consistent with the mutational data.A major concern is that there are too many genes studied at too shallow a depth. There are many lower resolution and correlative assays employed so that the conclusions rely on the aggregate acceptance of many different assays each with different alternative explanations, and some of which work better on some genes and selected elements than others.

Concerning "too many genes": The message of the paper is that several quadruplexes from different genes work cooperatively to regulate an important pathway. Nature often works in this fashion, so it is worthy of investigation. Such a study must consider the function of many elements in order to probe the pathway.

Concerning "too shallow": Briefly: from an initial screen of 35 elements (Figure 2), we followed up on 7 putative quadruplexes (PQS's), for which we acquired data using all known biophysical and cellular techniques for quadruplexes, including: (1) reporter assays + mutated controls in cells, (2) in vitro transcription/translation assays, (3) gel migrations, (4) ThioT stainings, (5) UV melts, (6) CD spectra (Figure 3). With the exception of ARG_3Q1_ and AZIN1 (for which we provided credible data-based explanations (Figure 2D, Figure 4, new Figure 5, Figure 5—figure supplement 1, Figure 5—figure supplement 3), the outcome from all of these techniques for the PQS's were consistent with our proposed structures and function. Finally, we investigated PQS's from AZIN 1 and SMS in cells (Figure 4). We showed that the endogenous proteins and luciferase reporters performed identically, in response to a known quadruplex-stabilizing drug and a polyamine depleting-drug with subsequent polyamine rescue. So, with respect, we see the referee's call for "stronger support" and "the need to accept this as an aggregate survey" as unjustifiable.

In the new manuscript we have investigated in more detail AZIN1 with a high-resolution NMR study. The data confirms our hypothesis that the AZIN1 quadruplex equilibrates with stem structures, and that the equilibrium is shifted in predictable fashion for a quadruplex by changing concentrations of potassium, magnesium and sodium.

The assays for structure determination aren't nucleotide resolution, and it remains tricky to predict these structures confidently by computational methods. Overall the need to accept this as an aggregate survey weakens the conclusiveness of the presentation. Having done this interesting survey, the authors are likely interested in higher resolution study of one or a few of these elements in detail that would really nail down the idea that G-quartets interact with polyamines to control expression of polyamine metabolic enzymes.

We are not sure that we understand the referee's comment. We have added a nucleotide-resolution assay with the ^1^H NMR study of AZIN_5Q1_ (new Figure 5 and new Figure 5—figure supplement 1).

Reviewer #2:In the manuscript, Lightfoot and Hall describe the effect of transient G_2_-quadruplexes on the expression levels of proteins involved in the polyamine biosynthesis pathway.Enzymes involved in this pathway are regulated at different levels including transcriptional and post-transcriptional levels.The authors hypothesized that these levels could be regulated at the post-transcriptional level by specific RNA structures, G-quadruplexes. Indeed, RNA G-quadruplexes have already been shown to play a role in RNA metabolism regulation, such as translation, splicing,.Using bioinformatic tools, the authors identified putative G-quadruplex forming sequences (PQS) in the 5' and 3'UTR of many genes involved in this pathway. They validated the effect of these putative G-quadruplexes using in vivo and in vitro assays on WT and mutated sequences, as well as the effect of a well-characterized G-quadruplex stabilizer.Overall, the manuscript is well written, clear, and sound. The authors nicely demonstrate that while many PQS do not have any effect on protein expression, other have a significant influence. The study of mutated sequences in itself is controversial (is the effect of the mutation due to a disruption of the G-quadruplex or a change in the ratio of proteins bound is always difficult to assess) but the complementary approach of testing the effect of a G-quadruplex stabilizer on protein expression makes the conclusions strong and compelling.The final conclusion of the manuscript that spermine itself stabilizes G-quadruplex is interesting but lacks in my view strong evidence. The in cells experiments cannot provide evidence that the observed effects are direct and the in vitro validation assays have been done only on one PQS (AZIN1).

The referee is referring to the final figure. We agree that the evidence for direct interaction is not strong and derives from in vitro experiments. The new NMR data does provide additional evidence for such an interaction, but as the direct interaction is not the main message of the paper, we have moved old Figure 5 to Figure 5—figure supplement 3 and have toned down (removed text and emphasized that data is from in vitro experiments) our implications of recognition of the AZIN1 quadruplex by spermine.

It would be interesting to test the effect of spermine of the stability of other PQS sequence that respond to spermine in cells such as SMS_5Q1_, SMS_3Q1_, SAT1_5Q1_ and OAZ2_5Q1_. It would also be interesting to compare the effect of spermine and PDS on G-quadruplex stabilization in vitro.

We did indeed test the effects of spermine on other G_2_-PQS's SMS and SAT1: there was a small effect on SMS_5Q1_ but no effect on SAT1. We added a statement on this to the manuscript (subsection “Predicted G_2_-PQS’s in PSP UTRs”). We did not perform experiments with PDS in vitro.

Subsection “PSP long-looped G_2_-PQSs perform as quadruplexes in cells”: The authors observed that the effect of PDS on AZIN1_5Q1_ was opposite to what was expected based on the mutant results. However, AZIN1 also has a PQS in its 3'UTR (AZIN1_3Q1_). It would be interesting to test the effect of PDS on this reporter.

Given the different status of the cells with respect to polyamine levels, the effect of PDS on AZIN1_5Q1_ was perhaps not so surprising. We have elaborated on this with a small additional text (subsection “PSP long-looped G_2_-PQSs perform as quadruplexes in cells”).

Reviewer #3:This is a well described study by Lightfoot and Hall that demonstrates many G-rich sequences regulate the RNA and protein levels of proteins important in polyamide (PA) biosynthesis. The authors identify many putative G-quadruplexes in the 5' and 3' UTRs of PA protein-coding genes and show that some of these elements can form G-quadruplex structures in vitro and provide evidence that they may function via these structures using a small molecule known to bind G-quadruplex structures. This work adds to our understanding of how these genes are regulated through potential G-quadruplex structures. Although this is an interesting body of work, several concerns reduce enthusiasm for publication in eLife.1) A major concern is that there doesn't appear to be any data in this manuscript or other publications (search by this reviewer of the literature) that show G-quadruplexes can form without a single nucleotide loop (usually defined as 1-7 nucleotides although longer loops exist as previously demonstrated and in this work). If these references exist it would be helpful for the authors to include. Structurally it doesn't seem possible for the four Gs to be split into two parts of the G-quad. This should be demonstrated or explained if previously published data hasn't demonstrated this unique G-quadruplex structure before. This point applies to the elements in SAT1, AZIN1 and OAZ3.

Piazza et al., 2017 shows quadruplexes with a zero-nucleotide loop. We have added a short sentence and reference in the text (subsection “Predicted G_2_-PQS’s in PSP UTRs”). We did not further characterize the zero-looped quadruplexes because this non-canonical structure is not the main message of the paper.

2) Presumably the authors tested if polyamines bind and stabilize / destabilize the putative G_2_ quadruplex structures. These results should be included or discussed because otherwise the reader is left wondering how the polyamines self-regulate except for the AZIN1 example.

We added a sentence to the text stating that polyamine addition had no effect on the UV melt traces of other long-looped G_2_-PQS (SMS and SAT1).

For the AZIN1 data, the authors should correct ΔAbs of ~0.2 to ~0.02 in the Results section (Long-looped AZIN1 G_2_-quadruplex is induced by PAs in vitro).

Corrected.

Again, please provide a rationale for how a G-quad can form without a loop nucleotide (see point #1) and not just state that high resolution structural studies will need to be performed (Discussion section). If the authors demonstrate G-quads can form without a loop this would be a major finding for the field and would be of great interest. The data for the 3 putative G-quads lacking loops is weak compared to the data provided for the G-quads with loops.3) A more minor concern is the data shown in Figure 2. Several of the gel images for the Thioflavin-T with the G-quadruplexes don't appear to match the quantitation shown in the next panel although this could be explained by loading differences. For example, for ODC1_5Q2_ there is a 2-fold difference in fluorescence quantitation, which doesn't seem to match the adjacent gel image. This might be explained by the significant difference in the loading of RNA seen in the first panel. If the RNA concentration levels or ability to detect RNA is significantly different for some species it may call into question the ability to detect differences between WT and mutant RNAs.

There are no differences in loading, which is done on the basis of OD measurement, and experiments are repeated multiple times. Apparent differences between wild type and mutants on the migration gels are likely because Syber Gold shows a degree of sequence-selective binding (as reported in Tuma et al., 1999).

4) A minor point is why do the authors not spend more time in the discussion on the most significant PA regulatory element they identified in SAT1 (30-fold change)?

Although SAT1 is interesting, we found AZIN1 more interesting due to its structural equilibria and its strong regulation in cells. We have therefore added results from an NMR study of AZIN1 to the revised manuscript (new Figure 5 and Figure 5—figure supplement 1).

[Editors' note: the author responses to the re-review follow.]

The reviewers have discussed the reviews with one another and the Reviewing Editor has drafted this decision to help you prepare a revised submission. It was generally agreed that the submission was much improved, but additional work was needed.Essential revisions:Given the very loose restraints on the G_2_ search criteria it is not surprising that many sequences were found. However, it seems possible that the G_2_s could be involved in alternative secondary structures that are not quadruplexes. Have the authors checked the PQS's for other secondary structures (for example, by M-fold?).

We have checked the principal PQS's of Figure 3 by M-Fold. As expected for G-rich seqs, most of them show possible hairpin structures. However, given the strong evidence for the PQS's shown in Figure 3, particularly from the UV_295_-melting and Thioflavin T staining we did not pursue other secondary structures.

Transfected dual luciferase assays can be highly variable and the gold standard would be to report errors from at least 3 independent replicates, not 2.

Our legend describing biological replicates for Figure 2A in the original manuscript was ambiguous: we have corrected it in the new manuscript. For each PQS that produced a statistically-significant difference between wild-type and control, the data represents 3-7 independent biological replicates. For PQS's that were inactive and were dropped from further investigation, data is presented from n>/=2 independent biological replicates.

For example, AZIN1 has a strong effect in the dual luciferase assay (Figure 2A) but does not behave like a quadruplex in the biophysical assays in Figure 3. This could be an indication that the dual luciferase assay is not necessarily reporting on quadruplex formation. Negative controls are needed showing that mutations in the predicted loop regions have no effect in the dual luciferase assay.

The referee's assumption that generally mutating the loop sequences of a quadruplex should have no effect on luciferase activity is incorrect, according to previously published work which states that loop sequence is one of the 4 major determinants of quadruplex stability (Pandey, Agarwala and Maiti, 2013). (We stated this in the Introduction "The stability of G-quadruplexes is governed by the number of G-quartets, the loop length and composition").

As an aside, we actually agree with the referee "that the dual luciferase assay is not necessarily reporting on quadruplex formation": strong data in our manuscript (especially the new NMR data) describes a quadruplex/hairpin equilibrium for AZIN1_5Q1_. Thus, the referee's suggested experiment would not be a negative control; the best negative control is mutation of key G's, as we have done.

AZIN1 showed no evidence for quadruplex formation in the biophysical assays (Figure 3), yet this is the only RNA that was chosen for further characterization by NMR in order to demonstrate quadruplex formation. The rationale for pursuing AZIN1 needs to be explained.

We explained the reasons for pursuing AZIN1 on Line 472 of the second submission. We have expanded on this a little more in the new submission (see subsection "Long-looped AZIN1 G_2_-quadruplex equilibrates with hairpin structures").

How does the addition of polyamines affect the biophysical characterizations in Figure 3?

The referee has not stated a purpose for this rather wide-ranging question, nor stated which quadruplexes he/she refers to. Does the referee mean to test ALL 14 RNAs of Figure 3 in presence of polyamines in gel migrations, ThioT stainings, UV-melts and CD? Testing one polyamine at one concentration on a quadruplex and its mutant after optimizing conditions (e.g. polyamine conc., the presence/absence of Mg/K), whilst avoiding RNA precipitation etc. represents dozens of experiments.

We respectfully submit that grossly expanding the data set in this way without a specified reason would not add useful support for quadruplex structures above what we already have shown with biophysical experiments of Figures 3 and the new exhaustive NMR investigation.

Figure 5. The ^1^H signals around 11 ppm are consistent with quadruplex formation but this is not conclusive as other types of base pairs can also give rise to resonances in this region of the proton spectrum. A ^1^H-^15^N through hydrogen bond NMR experiment is needed to really show this.

We agree with the referee and thank him for the suggestion. We performed these experiments, as well as additional experiments to nail this down.

Signals around 11 ppm are characteristic for G-imino protons involved in H-bonds in GU base pairs, or H-bonding interactions with O6 from guanines in G-quadruplexes (Jin, 1990; Wang et al., 1991; Wang et al., 1919; Smith and Feigon 1992). To confirm that our signals from AZIN1_wt_ at around 11 ppm (Figure 5b) were indeed G-imino protons, we in vitro-transcribed a ^15^N-labeled AZIN1_wt_ and recorded a ^1^H ^15^N-HSQC. Similarly to what was previously reported for RNA G-quadruplexes (Nasiri et al., 2016)^6^, we observed that the protons of AZIN1_wt_ located at 11 ppm and the quadruplex positive control (AZIN1_M2_) were bound to ^15^N atoms with a chemical shift of about 145 ppm (new Figure 5—figure supplement 2A,B, respectively), which is characteristic for guanine N1 atom involved in a G-quadruplex or a GU wobble.

Imino protons from GU base pairs can easily be seen in a ^1^H-^1^H 2D NOESY due to the strong OE between the imino protons of guanine and uridine; furthermore, U-imino protons are typically seen in a ^1^H ^15^N-HSQC at around 160 ppm for N3. Neither strong imino-imino NOE's, or H3-N3 imino cross-peaks, were observed in our spectra (new Figure 5—figure supplement 2). We therefore conclude that all the cross-peaks observed in the ^1^H ^15^N-HSQC at 10-11 ppm in the proton dimension originate from a G-quadruplex.

In addition, G-quadruplex formation also gives rise to typical NOE patterns involving the imino, amino and aromatic protons of the guanine nucleotides involved in the G-quadruplex formation (Smith and Feigon 2010; Jin, 1992 and Macaya, 1993). So, we then recorded a 2D NOESY with AZIN1_wt_ (new Figure 5—figure supplement 2C). NOEs' typical of G-quadruplexes between neighbouring imino groups and between imino and amino (shifted to 9-10ppm compared to Watson-Crick base-pairs) are present in the spectrum consistent with previous reports (Nasiri et al., 2016).

Taking these results together with the increased relative intensity of these imino signals in the presence of K (stabilizing the G-quadruplex) and the decrease in the presence of Na/Mg, provides strong evidence for formation of a G-quadruplex by AZIN1_wt_ (Figure 5c, Figure 5—figure supplement 1) The conclusion is strengthened by the 1D ^1^H and ^1^H ^15^N-HSQC spectra from the mutants AZIN1_M1_ and AZIN1_M2_ (new Figure 5—figure supplement 2B, Figure 5b).

Do the RNAs that respond to Thioflavin T also have similar 1D NMR spectra?

We prepared large quantities of ODC1_5Q1_, ARG2_5Q1_, OAZ2_5Q1_ and OAZ2_5Q2_ and tested their ability to form G-quadruplex structures by 1D NMR. We used NRAS as a positive control as it was previously shown to adopt a G-quadruplex structure (Bugaut et al., 2010). For all these RNAs, imino protons were detected at 11 ppm indicating their ability to form G-quadruplex structures (new Figure 5—figure supplement 1). We also prepared ^15^N-labeled versions of ARG2_5Q1_, OAZ2_5Q1_ and OAZ2_5Q2_. In each case, we found that the imino protons observed at 11 ppm were attached to nitrogen atoms with chemical shifts of about 145 ppm (new Figure 5—figure supplement 1E-G). We saw no evidence of GU base pairs formation for any of these RNAs.